# Neural space–time model for dynamic multi-shot imaging

Ruiming Cao ®[1] ✉, Nikita S. Divekar ®[2], James K. Nuñez ®[2], Srigokul Upadhyayula ®[2] & Laura Waller ®[3] ✉

Computational imaging reconstructions from multiple measurements that are captured sequentially often suffer from motion artifacts if the scene is dynamic. We propose a neural space–time model (NSTM) that jointly estimates the scene and its motion dynamics, without data priors or pre-training. Hence, we can both remove motion artifacts and resolve sample dynamics from the same set of raw measurements used for the conventional reconstruction. We demonstrate NSTM in three computational imaging systems: differential phase-contrast microscopy, three-dimensional structured illumination microscopy and rolling-shutter DiffuserCam. We show that NSTM can recover subcellular motion dynamics and thus reduce the misinterpretation of living systems caused by motion artifacts.

Multi-shot computational imaging systems capture multiple raw measurements sequentially and combine them through computational algorithms to reconstruct a final image that enhances the capabilities of the imaging system (for example, super-resolution[1,2], phase retrieval[3] and hyperspectral imaging[4]). Each raw measurement is captured under a different condition (for example, illumination coding and pupil coding) and hence encodes a different subset of the information. The reconstruction algorithm must then decode this information to generate the final reconstruction.

If the sample is moving during the multi-shot capture sequence, the reconstruction may be blurry or suffer artifacts[5] as the system effectively encodes information from slightly different scenes at each time point. Thus, most methods require that the sample be static during the full acquisition time, which limits the types of samples that can be imaged. Approaches for imaging dynamic samples aim to reduce acquisition time by multiplexing measurements via hardware modifications[6–8], developing more data-efficient reconstruction algorithms[9–11] or deploying additional data priors with deep-learning techniques[12–19]; however, these methods may be impractical to implement and usually are only applicable for a specific imaging system. Data priors, for example, are nontrivial to generate (for example, due to the lack of access to groundtruth data) and may fail with out-of-distribution samples[20].

Here we take another approach for imaging moving samples, where we model the sample dynamics to account for it during the image reconstruction. Modeling sample dynamics in multi-shot methods is challenging for two reasons. First, each measurement has a different encoding, so we cannot simply register the raw images to solve for the motion. Second, the motion can be highly complex and deformable, necessitating a pixel-level motion kernel. Our approach is to use deep-learning methods to develop flexible motion models that would be very difficult to express analytically. For example, recent work successfully used a deep-learning approach (with a robust data prior) to model dynamics in the case of single-molecule localization microscopy[21].

We propose a neural space–time model (NSTM) that can recover a dynamic scene by modeling its spatiotemporal relationship in multi-shot imaging reconstruction. NSTM exploits the temporal redundancy of dynamic scenes. This concept, widely used in video compression, assumes that a dynamic scene evolves smoothly over adjacent time points. Specifically, NSTM models a dynamic scene using two coordinate-based neural networks; these networks store the multi-dimensional signal through their network weights, and are used for novel view-synthesis[22], three-dimensional (3D) object representation[23] and image registration[24,25]. As illustrated in Fig. 1b, one network of NSTM represents the motion and the other network represents the scene. The motion network outputs a motion kernel for a given time point, which estimates the motion displacement for each pixel of the scene. Subsequently, the scene network generates

[1]Department of Bioengineering, UC Berkeley, Berkeley, CA, USA. [2]Department of Molecular and Cell Biology, UC Berkeley, Berkeley, CA, USA. [3]Department of Electrical Engineering and Computer Sciences, UC Berkeley, Berkeley, CA, USA. ✉e-mail: rcao@berkeley.edu; waller@berkeley.edu

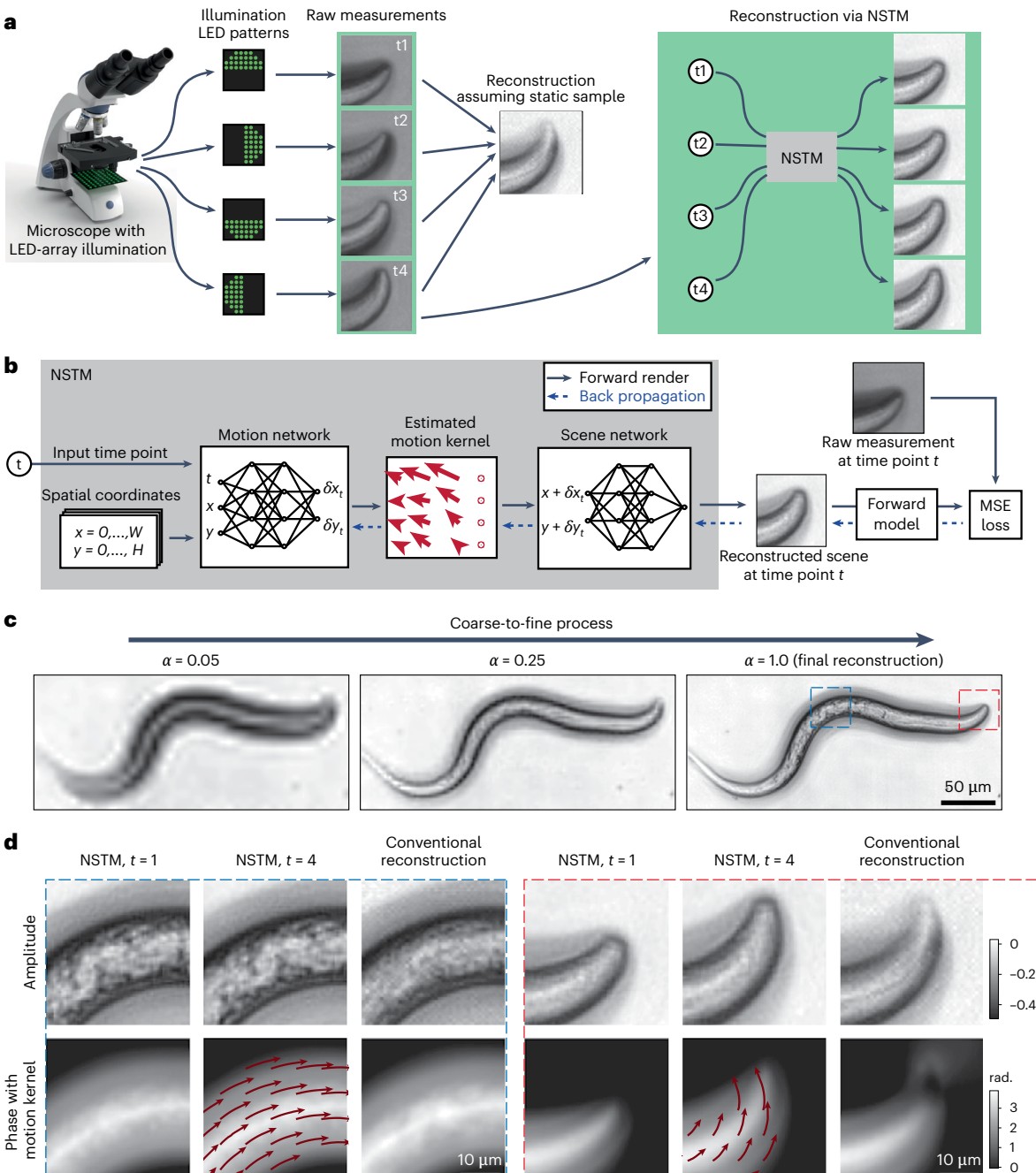

**Fig. 1 | The neural space–time model for dynamic imaging reconstruction.**
**a**, Multi-shot computational imaging systems capture a series of images under different conditions and then computationally reconstruct the final image. For example, DPC captures four images with different illumination source patterns, and then uses them to reconstruct quantitative phase. Sequential capture of the raw data results in motion artifacts for dynamic samples, as the reconstruction algorithm assumes a static scene. Our proposed NSTM extends such methods to dynamic scenes, by modeling and reconstructing the motion at each time point. **b**, NSTM consists of two coordinate-based neural networks, one for the motion and one for the scene. Once the networks have been trained using the dataset of raw measurements, we can give the NSTM any time point as the input and it will generate the reconstruction at that time point. The network weights of NSTM are trained to match the forward model-rendered measurement with the actual raw measurement at each time point. **c**, The coarse-to-fine process for the reconstruction of a live *C. elegans* worm imaged by DPC. **d**, Zoom-ins for NSTM reconstruction at different time points with the recovered motion kernel overlaid, along with a comparison to conventional reconstruction.

a scene using spatial coordinates that have been adjusted for motion by the motion network. Then, the generated scene is passed into the system's forward model to produce a rendered measurement. To train the weights of the two networks (which store the scene and its motion dynamics), we use gradient descent optimization to minimize the difference between the rendered measurements and the acquired measurements (Methods).

The motion and scene networks in NSTM are interdependent and failing to synchronize their updates leads to poor convergence of the model. This poor convergence typically happens when the scene network overfits to the measurements before the motion is recovered, a situation common for scenes involving more complex motion (Extended Data Figs. 1 and 2). To mitigate this issue, we developed a coarse-to-fine process (detailed in Methods), which controls the

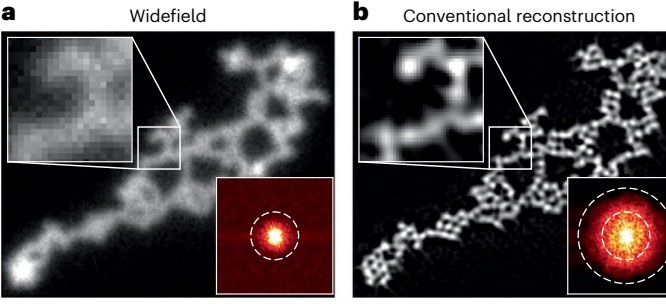
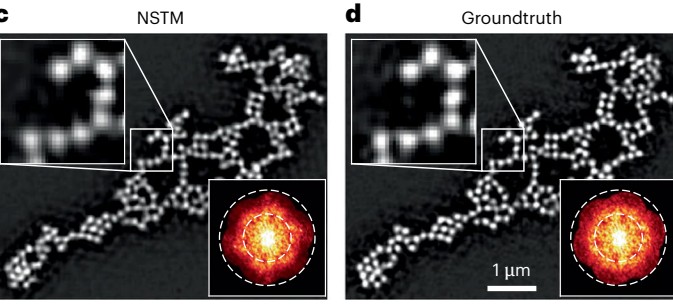

**Fig. 2 | Structured illumination microscopy of a dense microbead sample with vibrating motion. a**, The diffraction-limited widefield image cannot resolve individual beads. **b**, The conventional SIM reconstruction algorithm (fairSIM[31]) assumes a static scene, so suffers from motion blur. **c,d**, Our NSTM reconstruction resolves all of the subresolution-sized beads and gives a similar quality reconstruction (**c**) as the groundtruth case (**d**), in which we collected the data without sample motion. Bottom right of each image shows the frequency spectra (with gamma correction power of 0.7).

granularity of the outputs from both networks. Specifically, the reconstruction starts by recovering only the low-frequency features and motion and then gradually refines higher-frequency details and local deformable motion as illustrated in Fig. 1c.

NSTM is a general model for motion dynamics and can be plugged into any multi-shot system with a differentiable and deterministic forward model. It does not involve any pretraining or data priors; the learned network weights describe the final reconstructed video for each dataset individually, so it can be considered a type of 'self-supervised learning'. We demonstrate NSTM here for three different computational imaging systems: differential phase-contrast microscopy (DPC)[26], 3D structured illumination microscopy (SIM)[2] and rolling-shutter DiffuserCam[27]. In future, we hope it will find use in other applications as well.

## Results

### Differential phase-contrast microscopy

Our first multi-shot computational imaging system captures four raw images, from which it reconstructs the amplitude and phase of a sample[26]. The images are captured with four different illumination source patterns, which are generated by an LED array microscope in which the traditional brightfield illumination unit is replaced by a programmable LED array[28]. In Fig. 1a, we show the system and raw images captured for a live, moving *Caenorhabditis elegans* sample. The conventional reconstruction algorithm assumes a static scene over these four raw images. Consequently, unaccounted sample motion leads to artifacts in the reconstruction (Fig. 1d). Through the coarse-to-fine process (Fig. 1c), the NSTM recovers the motion of the *C. elegans* at each time point, giving both a clean reconstruction without motion artifacts and an estimate of the sample dynamics.

### 3D structured illumination microscopy

Our second multi-shot system is 3D SIM[2], which captures 15 raw measurements at each *z* plane (three illumination orientations and five phase shifts for each orientation). The conventional 3D SIM reconstruction assumes there is no motion during the acquisition; thus, it is limited to fixed samples. Previous work in extending 3D SIM to live cells focuses on accelerating the acquisition through faster hardware[8,29,30] or assumes translation-only motion[2]. NSTM provides a strategy to recover and account for deformable motion. Because we model motion during the acquisition of a single volume, we can reconstruct both the super-resolved image and the dynamics (Methods).

Figure 2 shows results for a single-layer dense microbead sample in which we introduced motion by gently pushing and releasing the optical table during the acquisition. Using a conventional reconstruction algorithm (fairSIM[31]) results in a motion blurred image in which the individual beads cannot be resolved. In contrast, our NSTM reconstruction resolves individual beads with a quality comparable to the

groundtruth reconstruction. In addition, we also recover the motion map (Extended Data Fig. 3b,d). In this experiment, the groundtruth was reconstructed from a separate set of raw measurements captured without motion (Fig. 2d).

Applying this technique to live-cell imaging, Fig. 3 and Extended Data Fig. 4 show 3D SIM reconstructions for a live RPE-1 cell expressing StayGold-tagged[32] mitochondrial matrix protein. In Fig. 3b, the conventional reconstruction seems to show a mitochondrion with a tubule branch (red arrow); however, our NSTM result recovers the sample dynamics (Extended Data Fig. 4b and Supplementary Video 3) and thus recognizes that it is actually a single tubule which is moving during the acquisition time. This can be further verified by the low-resolution widefield images (Fig. 3e) and by running our NSTM algorithm without the motion update (Extended Data Fig. 4c). In addition to resolving motion, NSTM removes motion blur, recovering features that were blurred in the conventional reconstruction (blue arrows in Fig. 3b,c) and thus NSTM preserves more high-frequency content compared to conventional reconstructions (Extended Data Fig. 4d).

In another 3D SIM experiment, we imaged a live RPE-1 cell expressing StayGold-tagged endoplasmic reticulum (ER) (Fig. 4). The conventional reconstruction struggles to resolve clear ER network structures, likely due to their fast dynamics (see red arrows). Additionally, the motion artifacts in the conventional reconstruction are changing over time, making it difficult to visually track different features to see the ER dynamics. NSTM, on the other hand, recovers the motion kernels and the dynamic scene from the same set of raw images for a single volume reconstruction and the ER structures that it resolves are consistent over time. The recovered motion kernels reveal the dynamics happening at different time points within a single 3D SIM acquisition as shown in Fig. 4c and Supplementary Video 4. We also imaged a live RPE-1 cell tagged with F-Actin Halo-JF585 to show NSTM's capability on dense subcellular structures (Extended Data Fig. 6 and Supplementary Video 5).

### Rolling-shutter DiffuserCam lensless imaging

Our third multi-shot computational imaging example is rolling-shutter DiffuserCam[27], a lensless camera that compressively encodes a high-speed video into a single captured image. This method leverages the fact that each row of the image, captured sequentially by the rolling shutter, contains information about the whole scene at that time point, due to the system's large point-spread-function (PSF). To enable video reconstruction from the single raw image, the original algorithm[27] uses total variation regularization to promote smoothness. In contrast, by modeling for the motion explicitly, NSTM produces cleaner reconstructions without over-smoothing (Extended Data Fig. 7b). As a byproduct of NSTM, the motion trajectory for any point can be queried directly from the motion network (Extended Data Fig. 7c).

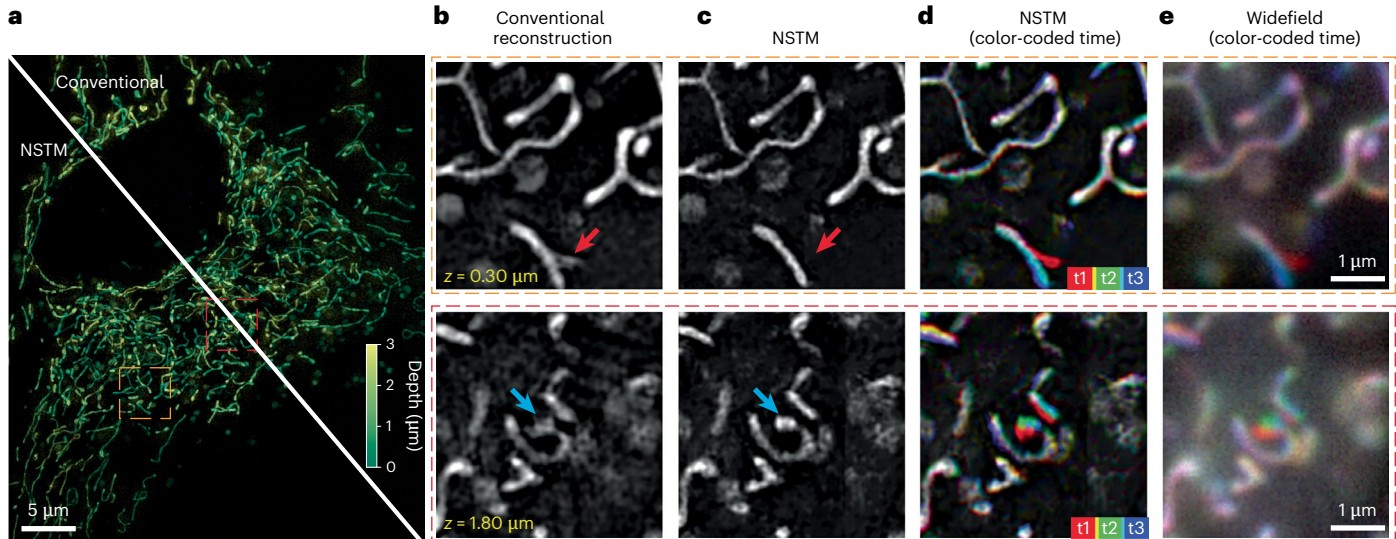

**Fig. 3 | 3D SIM reconstruction of a live RPE-1 cell expressing StayGold-tagged mitochondrial matrix protein. a,** Maximum projection of the volume with color-coded depth. **b,c,** Zoom-in of a slice from the 3D reconstruction, comparing the conventional 3D SIM algorithm (CUDA-accelerated three-beam SIM reconstruction software[2]) with our NSTM algorithm. The NSTM reconstruction disambiguates the artifacts induced by tubular motion (as indicated by the arrows). **d,e,** The NSTM reconstructions and widefield images at three timepoints coded by colors. Widefield images are obtained by summing the raw images from five phase shifts.

## Discussion

We demonstrated our NSTM for recovering motion dynamics and removing motion-induced artifacts in three different multi-shot imaging systems; however, the models are general and should find use in other multi-shot computational imaging methods. Notably, NSTM does not use any data priors or pretraining, such that the network weights are trained from scratch for each set of raw measurements. Hence, it is compatible with any multi-shot system with a differentiable and deterministic forward model. For multi-shot imaging systems such as 3D SIM, which do not use gradient-based reconstruction, we can alternatively implement a forward model as part of the NSTM reconstruction (Methods).

While NSTM is a powerful technique to resolve dynamic scenes from multiple raw images, it relies on temporal redundancy (the smoothness of motion and correlatable scenes over adjacent time points), to jointly recover the motion and the scene. As a consequence, this strategy tends to degrade or fail when the motion is less smooth. To demonstrate some failure modes, we provide several simulation examples. First, we simulate different amounts (magnitudes) of motion, showing that NSTM does well with large magnitudes of rigid-body or linear motion, presumably due to the effectiveness of coarse-to-fine process, but begins to degrade with large magnitudes of local deformable motion (Extended Data Fig. 8). Second, we simulate periodic local deformable motion with different vibration frequencies (Extended Data Fig. 9). We find that as NSTM does not explicitly account for periodic motion, it cannot capture high-frequency vibrations when the motion is no longer smooth between adjacent frames. Third, we simulate additive Gaussian noise to the raw measurements (Extended Data Fig. 10) to show how noise degrades the NSTM reconstruction.

One limitation of our method is that its two-network construction cannot accommodate for certain dynamics. Despite that this construction allows an explicit motion model and ensures reconstruction fidelity, it also introduces an additional constraint: as the scene network does not depend on the temporal coordinate, any frame of a dynamic scene has to be obtained by deforming a static reconstruction (from the scene network) with a motion kernel (from the motion network). As a result, NSTM is unable to recover dynamic scenes with appearing/disappearing features or switching on/off dynamics (such

as neuron firing or fluorescence photoactivation), which cannot be reproduced by a time-independent scene network. To overcome this limit, future work could modify the NSTM architecture to account for the different types of nonsmooth dynamics and/or incorporate the time-dependency to the scene network.

Another limitation is that our NSTM reconstructions generally require more computation than conventional methods. For example, the dense microbead reconstruction using NSTM took ~3 min on a NVIDIA RTX 3090 GPU, in contrast to the conventional algorithm (fairSIM) which completed in less than 10 s on a CPU. The live-cell 3D reconstructions (volume size 20 × 512 × 512 with 15 time points) using NSTM took 40.5 min on a NVIDIA A6000 GPU (Supplementary Table 1). Future work could improve the computational efficiency of NSTM by better initialization of network weights[33], hyper-parameter search for a faster convergence[34], using lower precision arithmetic and data-driven methods to optimize a part of the model in a single pass[35].

One interesting advantage of using coordinate-based neural networks like NSTM is that it can accommodate arbitrary coordinates that may not be on a rectilinear grid. This is especially advantageous for modeling spatiotemporal relationships, as it can intuitively handle sub-pixel motion shifts and nonuniformly sampled measurements in both space and time, without requiring interpolation of a uniformly sampled matrix. For example, one can output a temporally interpolated video with any desired temporal resolution simply by querying the network at intermediate time points between actual measurement timepoints to render the corresponding frames, as demonstrated in Supplementary Video 6. The resulting reconstructions are clean (no motion blur) and can faithfully represent the scene at those timepoints, provided that the dynamics are accurately modeled by the NSTM. We should not, however, expect to recover any dynamics happening at timescales faster than those that can be learned from the measurements.

In summary, we showed that our NSTM method can recover motion dynamics and thus resolve motion artifacts in multi-shot computational imaging systems, using only the typical datasets used for conventional reconstructions. The ability to recover dynamic samples within a single multi-shot acquisition seems particularly promising for observing subcellular systems in live biological samples. By accounting for motion through NSTM's joint reconstruction, NSTM reduces the risk

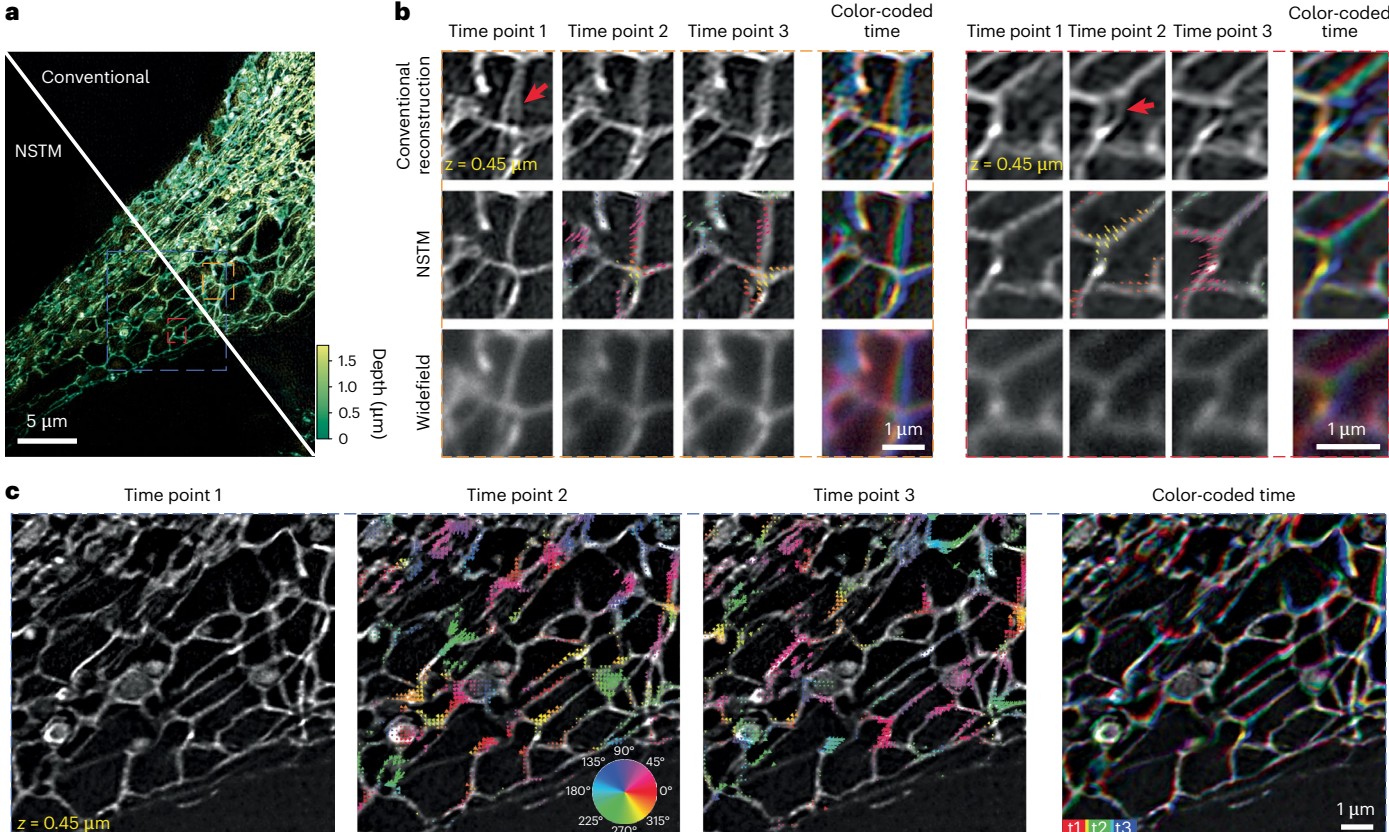

**Fig. 4 | 3D SIM experimental results for a RPE-1 cell expressing StayGold-tagged endoplasmic reticulum. a**, Maximum projection of the reconstructed volume with color-coded depth. **b**, Zoom-ins at three time points, each of which corresponds to a different illumination orientation, for the widefield, conventional and NSTM reconstructions. A moving window approach (Methods) is used to compute the conventional reconstruction[2] at different time points. The NSTM reconstructions are overlaid with the recovered motion kernels which show the sample's motion displacements from the previous time point. The colors of the motion kernel indicate motion directions, according to the colorwheel in **c**. **c**, Zoom-in NSTM reconstructions at three time points and the combined view with color-coded time. The motion kernels on the second and third time points show the structure's motion displacements from the previous time point, with color-coding to indicate motion directions.

of misinterpretations in the study of living systems caused by motion artifacts in multi-shot acquisitions. Further, it effectively increases the temporal resolution of the system when multi-shot data are captured.

## Online content

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

## Methods

### Cell-line generation

The RPE-1 cell lines used in 3D SIM experiments were cultured using Dulbecco's modified Eagle medium/Nutrient Mixture F12 (Thermo Scientific, 11320033) supplemented with 10% FBS (VWR Life Science 100% Mexico Origin 156B19), 2 mM L-glutamine, 100 U ml⁻¹ penicillin and 100 mg ml⁻¹ streptomycin (Fisher Scientific, 10378016). Trypsin–EDTA (0.25%) phenol red (Fisher Scientific, 25200114) was used to detach cells for passaging. To generate the cell lines, we obtained the pCSII-EF/mt-(n1)StayGold (Addgene, plasmid #185823) and pcDNA3/er-(n2)oxStayGold(c4)v2.0 (Addgene, plasmid #186296) from A. Miyawaki[32] to tag the mitochondrial matrix and the ER, respectively. We obtained the LifeAct-HaloTag from D. Gadella (Addgene #176105) to tag F-Actin. The er-(n2)oxStayGold(cr)v2.0, mt-(n1)StayGold and the LifeAct-HaloTag sequences were PCR amplified and cloned into a lentiviral vector containing an EF1α promoter. The vector is a derivative of Addgene #60955 with the sgRNA sequence removed. Lentiviral particles containing each plasmid were produced by transfecting standard packaging vectors along with the plasmids into HEK293T cells (ATCC CLR-3216) using TransIT-LT1 Transfection Reagent (Mirus, MIR2306). The medium was changed 24 h post-transfection without disturbing the adhered cells and the viral supernatant was collected approximately 50 h post-transfection. The supernatant was filtered through a 0.45-mm PVDF syringe filter and ~1 ml was used to directly seed a 10-cm plate of hTERT RPE-1 cells (ATCC CRL-4000). Two days post-infection, cells were analyzed on BD FACSAria Fusion Sorter and BD FACSDiva Software. The highest 5% of StayGold/GFP (FITC) fluorescence cells were sorted for the StayGold-tagged-ER and mitochondrial matrix lines (gating strategy illustrated in Supplementary Fig. 1). To prepare F-Actin Halo-tagged RPE-1 cells for sorting, Janelia Fluor HaloTag Ligand 503 was diluted at 1:20,000 from a 1 mM stock in supplemented DMEM-F12. Then the original medium was carefully aspirated off the cells and replaced with DMEM-F12 medium containing the ligand. The ligand and cells were incubated at 37 °C for 15 min, then washed three times with PBS before trypsinization and subsequent sorting. For the LifeAct-Halo-tagged RPE-1 line, the same gating strategy was used as described above for StayGold cells wherein highest 5% of Halo fluorescence cells were sorted (gating strategy illustrated in Supplementary Fig. 2). All sorted cells were expanded for imaging experiments.

### Sample preparation

Janelia Fluor JF585 dye was used to label the F-Actin on the LifeAct-Halo-tagged RPE-1 cells before imaging. The dense microbead sample was made with 0.19-μm dyed microbeads (Bangs Laboratories, FC02F). The stock solution was diluted 1:100 with distilled water and placed on a glass-bottom 35-mm dish coated by poly-L-lysine solution (Sigma Aldrich, P8920).

### Data acquisition

The 3D SIM datasets were acquired on a commercial three-beam SIM system (Zeiss Elyra PS.1) using an oil immersion objective (Zeiss, ×100 1.46 NA) and ×1.6 tube lens. The effective pixel size was 40.6 nm. The system captures 15 images at each depth plane, with three illumination orientations and five phase shifts for each orientation. A single image plane was acquired for the dense microbead sample. Twenty planes with a step size of 150 nm were captured for the RPE-1 cell expressing StayGold-tagged mitochondrial matrix protein, LifeAct-Halo-tagged RPE-1 cell stained with Janelia Fluor JF585 and 12 planes with a step size of 150 nm were captured for the RPE-1 cell expressing StayGold-tagged ER. A 488 nm laser was used for all but the F-Actin Halo-JF585 tagged cell, for which we used a 561 nm laser. The SIM system has a illumination update delay of around 20 ms for each phase shift or z-position shift, and a delay of 300 ms for each illumination orientation change. We set the exposure time to 20 ms for the dense microbeads and 5 ms for all cell experiments.

The DPC images were obtained from[36] with a commercial inverted microscope (Nikon TE300) with ×10 0.25 NA objective (Nikon) and an effective pixel size of 0.454 μm. An LED array[28] (SCI Microscopy) was attached to the microscope in place of the Köhler illumination unit. Four half-circular illumination patterns, with the maximum illumination NA equal to the objective NA, were sequentially displayed on the LED array to capture four raw images[26]. The exposure time was 25 ms.

The rolling-shutter DiffuserCam data are from the original work on the technique[27]. The raw image was taken by a color sCMOS (PCO Edge 5.5) in slow-scan rolling-shutter mode (27.52 μs readout time for each row) with dual shutter readout and 1,320 μs exposure time. The acquisition of the raw image took 31.0 ms.

### Construction of NSTM

The motion and the scene network of NSTM are both coordinate-based neural networks[22,23,37], a type of multi-layer perceptrons that learn a mapping from coordinates to signals. A coordinate-based neural network can represent a multi-dimensional signal, for example, an image or a 3D scene, through its network weights. To enhance the capacity and efficiency of the coordinate-based networks, we use hash embedding[38] to store multiple grids of features at different resolutions and transform a coordinate vector to a multi-resolution hash-embedded feature vector, $h = [h_0, h_1, \cdots, h_{N-1}]$, before passing it into the network (details in Supplementary Text on Hash Embedding). As the input coordinate varies, a fine resolution feature (for example, $h_{N-1}$) changes more rapidly than a coarse resolution feature (for example, $h_0$). During the coarse-to-fine process, we re-weight the output features of the hash embedding using a granularity value, $\alpha$, to control the granularity of the network. $\alpha$ is set by the ratio of the current epoch to the end epoch of the coarse-to-fine process, which is set to 80% of the total number of reconstruction epochs in practice. As in ref. 39, each feature $f_i$ is weighted by $\frac{1}{2} - \frac{1}{2} \cos(\pi \operatorname{trunc}(\alpha N - i))$, where trunc truncates a value to $[0, 1]$. In this way, finer features will be weighted to 0 until $\alpha$ gets larger, as illustrated in Fig. 1c.

In the forward process of NSTM (Fig. 1b), every spatial coordinate of the scene, $x$, is concatenated with the temporal coordinate $t$, and the hash-embedded features of the spatiotemporal coordinate, $hash(x, t)$, are fed into the motion network. The motion network, $f(\theta_{\text{motion}})$, produces the estimated motion displacement vector, $\delta x$, for each input spatiotemporal coordinate:

$$\delta x = f(hash(x, t) | \theta_{\text{motion}}). \tag{1}$$

The motion-adjusted spatial coordinate, $(x + \delta x)$ is then transformed into hash-embedded features and fed into the scene network, $f(\cdot | \theta_{\text{scene}})$ for the reconstruction value, $o$, such that

$$o(x, t) = f(hash(x + \delta x) | \theta_{\text{scene}}). \tag{2}$$

This process is repeated for all spatial coordinates to obtain the reconstructed scene at time $t$. As the scene network does not take the time as an input, it relies on the motion network to generate a dynamic scene. In our demonstrations, the scene network outputs a single channel as the fluorescent density for 3D SIM, two channels as the amplitude and phase for DPC and three channels as RGB intensity for DiffuserCam. As the hash embedding is always applied to the network input coordinate, we consider it a part of the network, $f$, and drop it from our expression for readability.

### NSTM reconstruction

To update the network weights of NSTM, the reconstructed scene is passed into the imaging system's forward model for a rendered measurement. Comparing the rendered measurement with the actual measurement acquired in the experiment, we compute the mean square

error loss and minimize it by back-propagating its gradient to update the network weights. Mathematically, the optimization becomes

$$\underset{\theta_{\text{motion}}, \theta_{\text{scene}}}{\arg\min} \sum_{i \in \{0, \cdots, T-1\}} \left(\text{forward}_i \left(f(\boldsymbol{x} + f(\boldsymbol{x}, t_i) | \theta_{\text{motion}}) | \theta_{\text{scene}}\right) - I_i\right)^2, \quad (3)$$

where $\text{forward}_i$ is the forward model to render the $i$th measurement given the temporal coordinate $t_i$. The actual measurement captured at time point $t_i$ is denoted as $I_i$. Adapting NSTM to new computational imaging modalities thus amounts to simply dropping in the appropriate forward model.

In our implementation, the motion network has two hidden layers with a width of 32 and the scene network has two hidden layers with a width of 128. The gradient update is performed with Adam optimizer[40]. The initial learning rate is set to $1 \times 10^{-5}$ for motion network ($5 \times 10^{-5}$ for rolling-shutter DiffuserCam reconstruction) and $1 \times 10^{-3}$ for scene network, with a exponential decay schedule to a tenth of the initial learning rate at the end of the reconstruction. For the conventional reconstruction of NSTM without motion update (in Extended Data Fig. 3a and Extended Data Figs. 4c and 5b), we keep all settings the same as the NSTM reconstruction except that the motion network is not updated and the input time points are set to zero. The NSTM reconstruction is implemented using Python and JAX[41].

## DPC reconstruction

The raw images of DPC are normalized by the background intensity and then passed through the linear transfer functions derived in ref. 26 as the forward model:

$$\text{forward}_i(o_u, o_p) = \mathcal{F}_{2D}^{-1}\left[H_u^i \mathcal{F}_{2D}(o_u) + H_p^i \mathcal{F}_{2D}(o_p)\right], \quad (4)$$

where $\mathcal{F}_{2D}$ is two-dimensional (2D) Fourier transform, $H_u^i$, $H_p^i$ denote the absorption and phase transfer functions for the $i$th measurement, and $o_u$ and $o_p$ are the absorption and quantitative phase of the scene. The conventional reconstruction is obtained by solving a Tikhonov regularization with a regularization weight of $10^{-4}$ for both amplitude and phase terms[26]. For ease of comparison, we add the same Tikhonov regularization to the loss term for NSTM reconstruction.

## 3D SIM reconstruction

The conventional 3D SIM reconstruction uses five measurements of different sinusoidal phase shifts to separate the complex spectra of three frequency bands and then shifts each band accordingly based on its corresponding modulation frequency. The band separation process necessitates the assumption of a static scene over those five measurements. To avoid this static assumption and preserve the temporal information, we implement the 3D SIM forward model in real space without band separation, rendering each measurement independently from NSTM's reconstruction at the time point that the actual measurement is taken.

This forward model can be expressed mathematically as

$$\text{forward}_i(o) = \sum_{j \in \{0,1,2\}} \mathcal{F}_{3D}^{-1}\left[\text{OTF}_j \, \mathcal{F}_{3D}(\text{illum}_{i,j} \, o)\right], \quad (5)$$

where $\mathcal{F}_{3D}$ denotes 3D Fourier transform. The super-resolved 3D fluorescent density, $o$, is first modulated by the corresponding illumination pattern, $\text{illum}_{i,j}$, at the $i$th measurement and band $j$. Then, the modulated signal is filtered by the optical transfer function, $\text{OTF}_j$, for each band $j$ and the resulted signals for the three bands are summed to render the $i$th intensity measurement.

In the naive implementation, we need to feed the 3D fluorescent density, $o$, at hundreds of different time points to the forward model to render a set of measurements, which is computationally inefficient.

For example, a dataset with 20 depth planes has 20 planes × 3 orientations × 5 phases = 300 raw images that contain 300 distinct time points. To improve the efficiency, we group together measurements with identical orientation and phase captured at different depth planes and render them in one forward model pass as if they were acquired at the same time point. This simple modification allows us to feed $o$ at only 15 time points to get the full set of raw images, regardless of the number of depth planes.

In our comparisons, we use the same illumination parameters estimated from measurements[2,29] for both conventional reconstruction algorithms and NSTM. For the conventional reconstructions shown in Fig. 4b, we use the moving window approach to select a set of raw images around a certain time point to feed into the reconstruction algorithm and we repeat this process to get the conventional reconstruction at every illumination orientation. For example, the conventional reconstruction at time point 3 in Fig. 4b uses raw images from illumination orientation 2 and 3 from the current acquisition and also the illumination orientation 1 from the next acquisition, where there is no delay between two acquisitions. Note that the term 'acquisition' here refers to 'time point' in a regular context of time-series acquisition, as 'time point' is already heavily used for time within a single acquisition of a scene.

## Rolling-shutter DiffuserCam reconstruction

Each row of the raw image captured by rolling-shutter DiffuserCam is the time integral of the dynamic scene convolved with the caustic PSF over the rolling-shutter exposure. Thus, its forward model can be written in a discrete-time sum of $T$ time points[27],

$$\text{forward}(o) = \sum_{t=0}^{T-1} (o(t) * \text{PSF}) \, S(t), \quad (6)$$

where $o$ is the dynamic scene, $S$ is a binary map of the shutter on/off state and * denotes 2D convolution operation. However, rendering the entire image at once requires obtaining NSTM's reconstructed scenes at all time points, which will be intensive on GPU memory. To make this feasible on common GPUs, during each step of the reconstruction we render a subset of image rows by only obtaining the reconstructed scenes at time points which have contributed signal to these rows. The forward model for the $i$th row of the raw image can be written as

$$\text{forward}_i(o) = \sum_{t \in \{t | S(i,t)=1\}} (o(t) * \text{PSF}) \, S(t). \quad (7)$$

In practice, to improve the efficiency, we render 20 consecutive rows in each forward pass.

## Reproducibility

The microbead with vibrating motion experiment shown in Fig. 2 and Extended Data Fig. 3 was repeated nine times. The optical table was pushed and released each time. Seven out of nine acquired datasets were suitable for NSTM reconstruction and produced similar results. The remaining two datasets suffered from severe motion blur in individual raw images and, thus, could not be recovered by NSTM.

## Reporting summary

Further information on research design is available in the Nature Portfolio Reporting Summary linked to this article.

## Data availability

SIM datasets collected in this study were deposited in Zenodo at https://doi.org/10.5281/zenodo.13204660 (ref. 42). DPC and rolling-shutter DiffuserCam datasets were obtained from refs. 27,36 and are also available at https://github.com/rmcao/nstm.

## Code availability

NSTM software is available at https://github.com/rmcao/nstm.

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

## Acknowledgements

We thank M. Kellman for sharing the DPC data and N. Antipa for sharing the rolling-shutter DiffuserCam data. This work was supported by the Weill Neurohub Investigators Program to L.W., CZI grant DAF2021-225666 and a grant from the Chan Zuckerberg Initiative DAF to L.W. (https://doi.org/10.37921/192752jrgbnh), an advised fund of Silicon Valley Community Foundation (funder https://doi.org/10.13039/100014989) to L.W., STROBE: A National Science Foundation Science & Technology Center under grant no. DMR 1548924 (NSF grant 1351896) to L.W., Chan Zuckerberg Biohub – San Francisco Investigators program to J.K.N., S.U. and L.W., Siebel Scholars program to R.C., CIRM training program EDUC4-12790 to N.S.D., Hanna Gray Fellowship from the Howard Hughes Medical Institute to J.K.N., Philomathia Foundation to S.U., Chan Zuckerberg Initiative Imaging Scientist program to S.U. and Lawrence Berkeley National Lab's LDRD to S.U. The SIM microscope facility was supported in part by the National Institutes of Health S10 program under award no. 1S10OD018136-01. The content is solely the responsibility of the authors and does not necessarily represent the official views of the National Institutes of Health.

## Author contributions

R.C., S.U. and L.W. conceived the work. R.C. developed the method and performed the experiments. S.U. and L.W. supervised this study. N.S.D. and J.K.N. generated the cell lines. R.C., N.S.D., J.K.N., S.U. and L.W. wrote the manuscript.

## Competing interests

L.W. has a financial interest in SCI Microscopy. R.C., N.S.D., J.K.N. and S.U. declare no competing interests.

## Additional information

**Extended data** is available for this paper at https://doi.org/10.1038/s41592-024-02417-0.

**Correspondence and requests for materials** should be addressed to Ruiming Cao or Laura Waller.

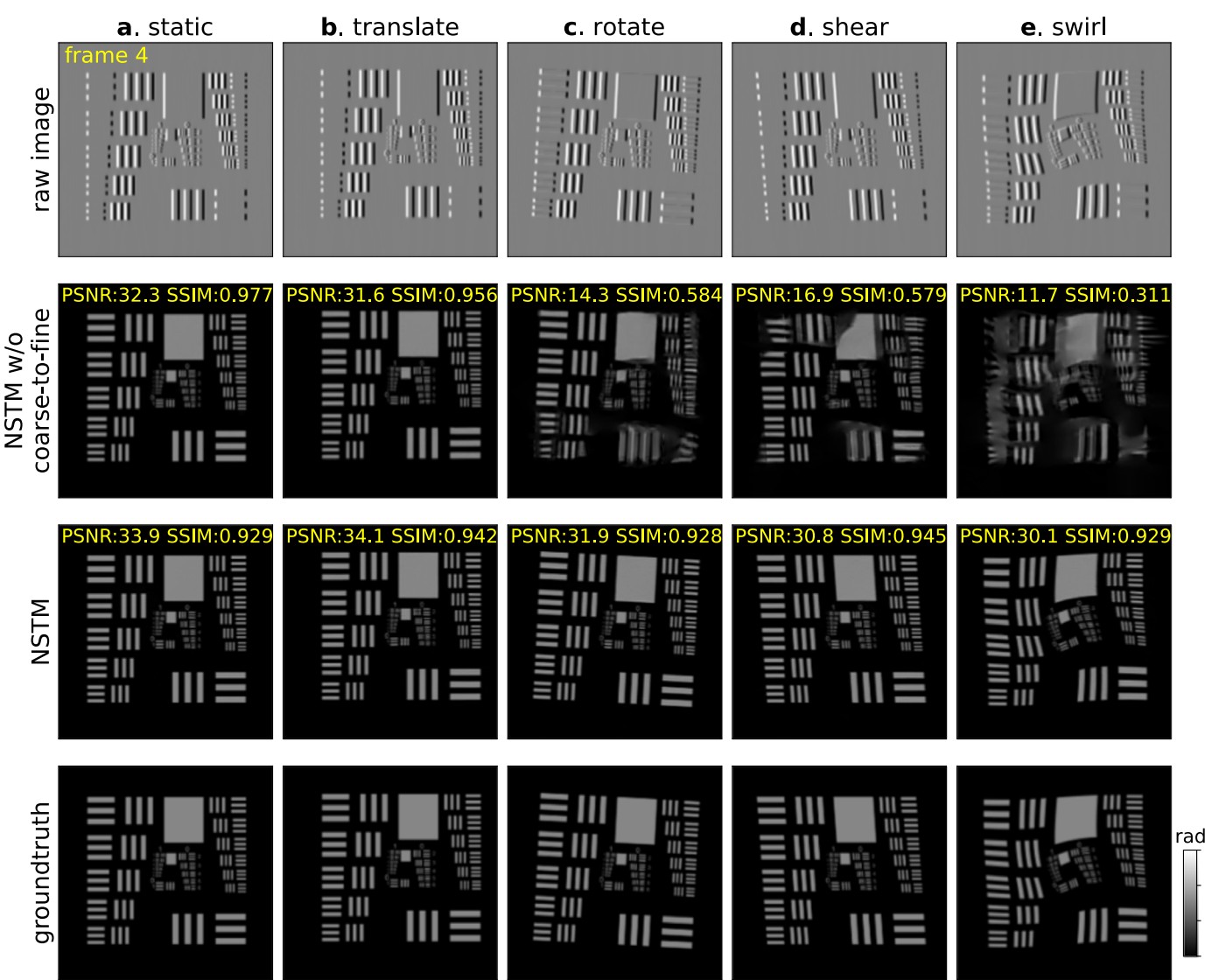

**Extended Data Fig. 1 | Simulations of differential phase contrast microscopy (DPC) using a phase-only USAF-1951 resolution target with various types of motion. a**, no motion, **b**, rigid motion - translation, **c**, rigid motion - rotation, **d**, non-rigid global motion - shearing, and **e**, local deformable motion - swirl. We reconstruct the quantitative phase of the dynamic scene using NSTM with the set of four simulated DPC images. Two reconstruction quality metrics are calculated: peak signal-to-noise ratio (PSNR) and the structural similarity index measure (SSIM). The NSTM does well with all types of motion. However, without using our coarse-to-fine process ('NSTM w/o coarse-to-fine'), it is likely to fail as the motion gets complicated, due to poor convergence of the joint optimization of motion and scene. Full videos of the dynamic reconstructions can be seen in Supplementary Video 1.

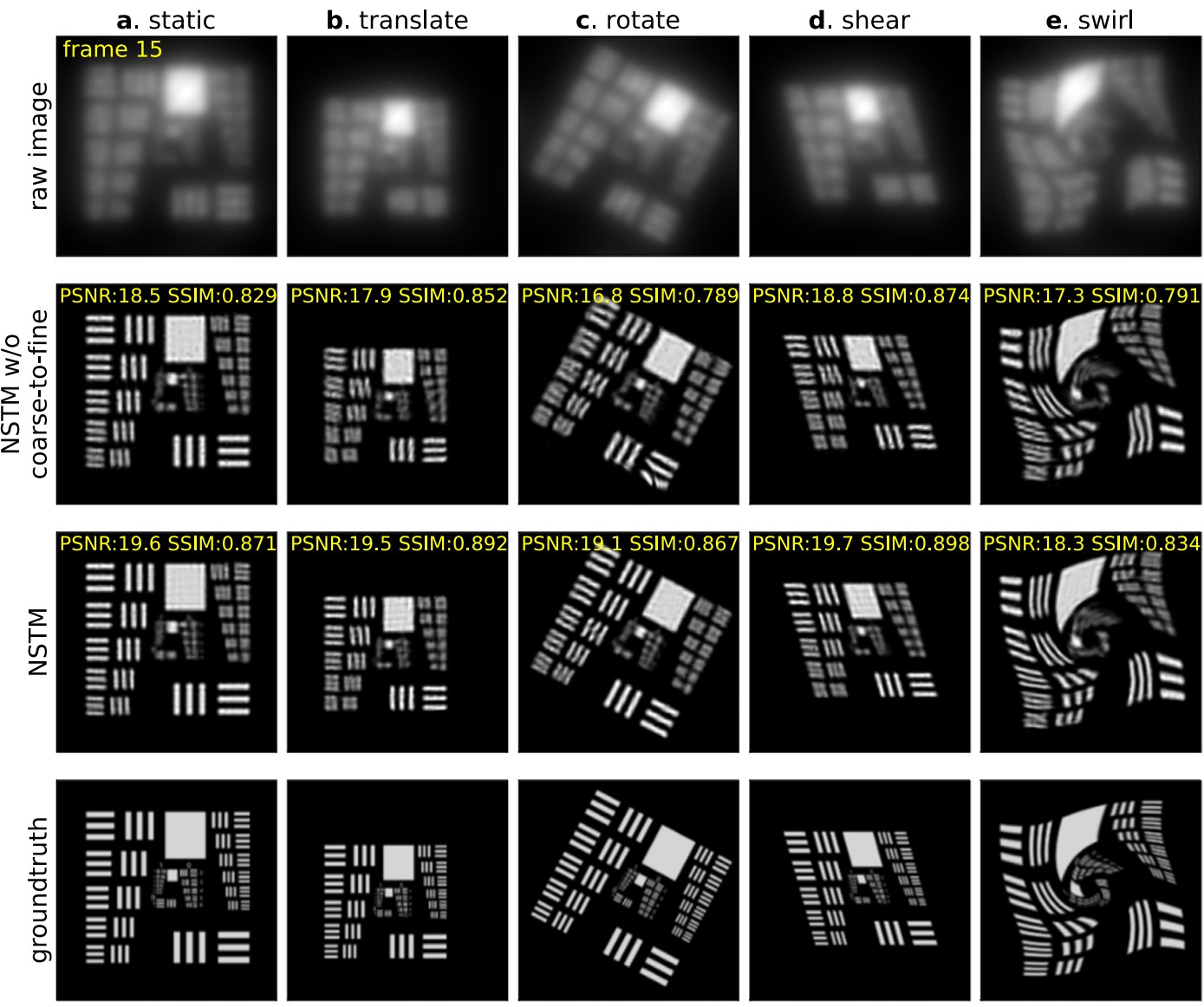

**Extended Data Fig. 2 | Simulations of structured illumination microscopy (SIM) using fluorescent USAF-1951 resolution target with various types of motion. a**, no motion, **b**, rigid motion - translation, **c**, rigid motion - rotation, **d**, non-rigid global motion - shearing, and **e**, local deformable motion - swirl. The forward model of single-plane three-beam SIM is assumed for the simulation. Full videos of the dynamic reconstruction can be seen in Supplementary Video 2.

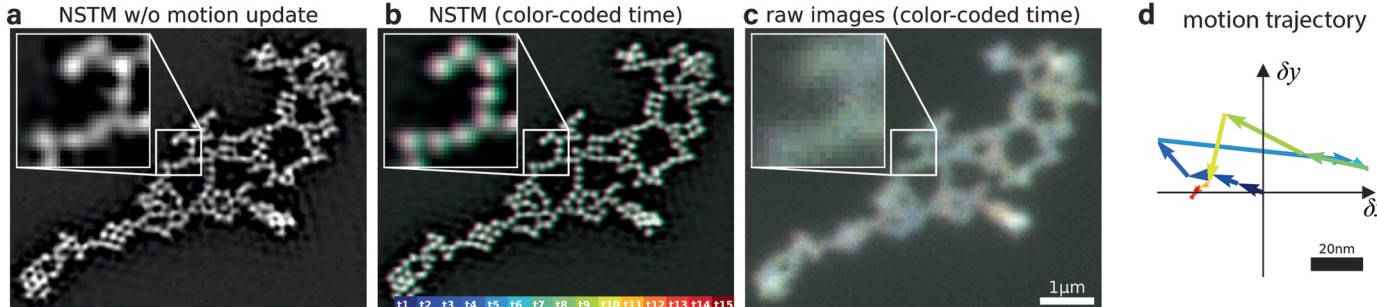

**Extended Data Fig. 3 | Additional results for the dense microbead sample from Fig. 2. a**, Reconstruction using NSTM without the motion update results in motion blurring similar to the conventional reconstruction in Fig. 2b, since dynamics are not accounted for. **b**, NSTM reconstruction with color-coded time. **c**, The raw images with color-coded time. In the images with color-coded time, each timepoint of raw images or reconstruction is drawn in a distinct color as indicated by the color bar. The 'color dispersion' in the zoom-in reconstruction suggests that subtle motion is recovered by NSTM. **d**, The recovered motion trajectory of a pixel on the vibrating microbeads from NSTM reconstruction. Each arrow shows the motion displacement vector with respect to the previous timepoint as indicated by the color code (color bar in **b**).

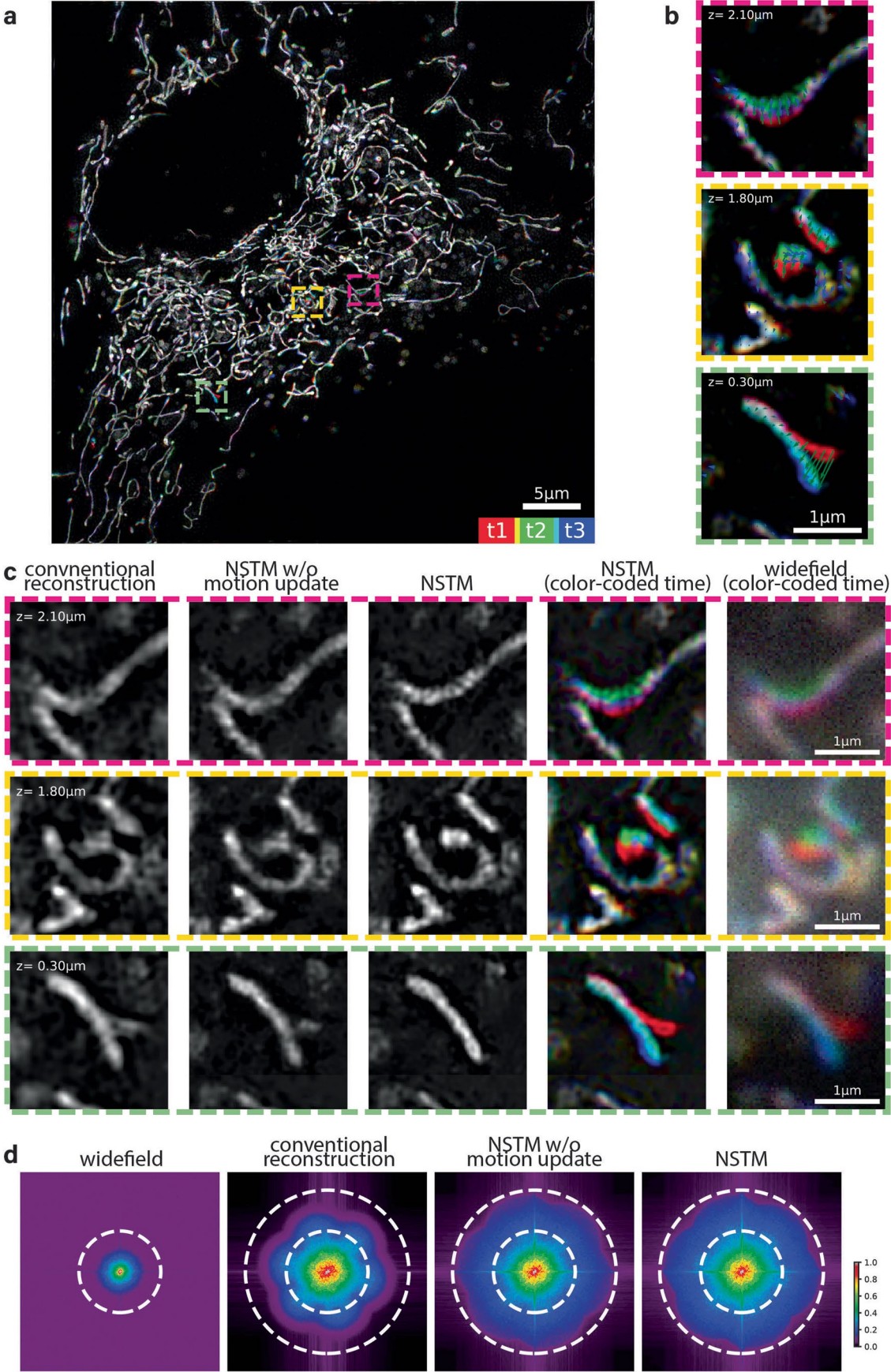

**Extended Data Fig. 4 | See next page for caption.**

**Extended Data Fig. 4 | Additional 3D SIM results for the mitochondria-labeled RPE-1 cell from Fig. 3. a**, Maximum projection of NSTM reconstruction volume, with three colors denoting the three timepoints that correspond to the three illumination orientations. **b**, Zoom-ins of a slice of NSTM 3D reconstruction, with color-coded time. The overlaid vector fields show the motion displacement recovered by NSTM, with their colors to indicate their corresponding timepoints. **c**, Zoom-in comparisons, from left to right: conventional reconstructions[2],

NSTM without motion update, NSTM reconstruction, NSTM reconstruction with color-coded time (three colors for three illumination orientations), and widefield images with color-coded time. **d**, A comparison of the spatial frequency spectra for each method. The two dashed circles indicate the diffraction-limited bandwidth and SIM super-resolved bandwidth, respectively. Gamma correction with power of 0.5 is applied to all frequency spectra for better contrast.

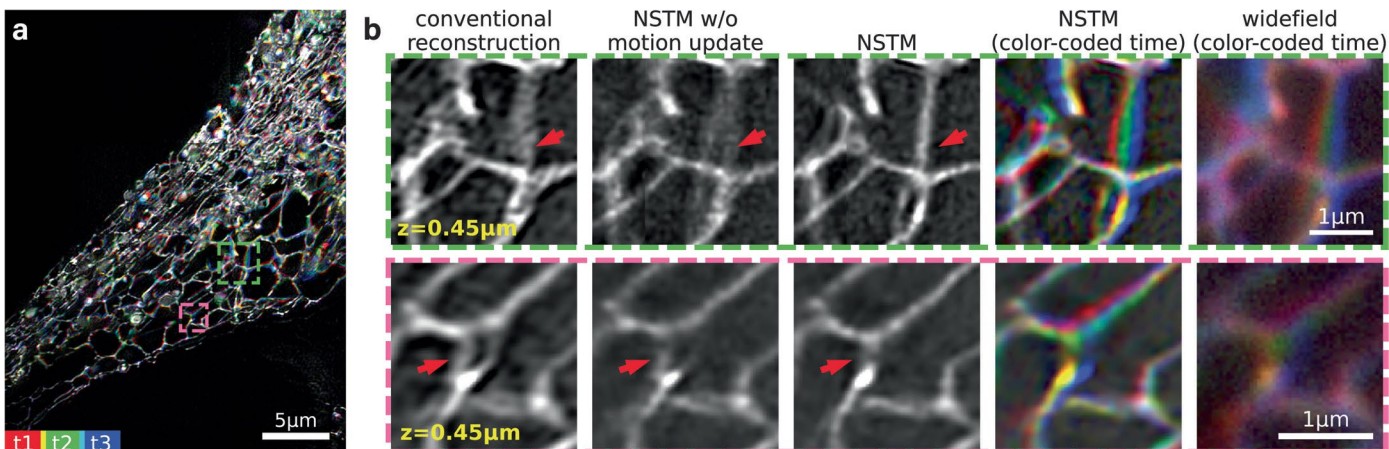

**Extended Data Fig. 5 | Additional 3D SIM results for the live endoplasmic reticulum-labeled RPE-1 cell from Fig. 4. a**, Maximum z-projection of NSTM reconstruction volume, with three colors denoting the three timepoints that correspond to the three illumination orientations. **b**, Zoom-in comparisons, from left to right: conventional reconstructions[2], NSTM without motion update, NSTM reconstruction, NSTM reconstruction with color-coded time (three colors for three illumination orientations), and widefield images with color-coded time.

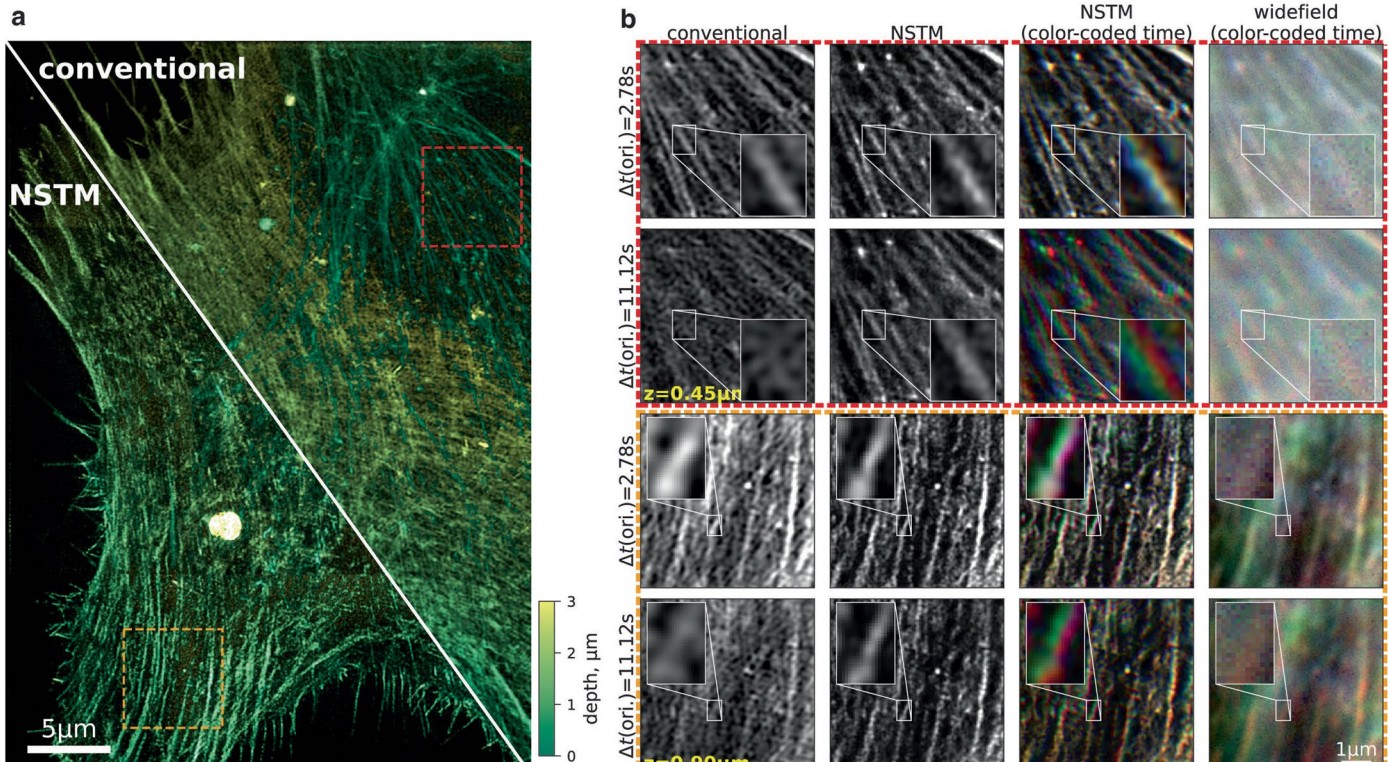

**Extended Data Fig. 6 | 3D SIM reconstruction of a live F-Actin labeled RPE-1 cell. a**, Maximum z-projection of the reconstructed volume with color-coded depth. **b**, Zoom-in comparisons, from left to right: conventional reconstruction[2], NSTM reconstruction, NSTM reconstruction with color-coded time (three colors for three illumination orientations), and widefield images with color-coded time.

The second row of each zoom-in assumes raw images with longer delay between orientations, $\Delta t$(ori.), and thus more motion (*that is*, the raw images of orientation 1 are from acquisition timepoint 1, orientation 2 from acquisition timepoint 2, and orientation 3 from timepoint 3 from a time-series measurement).

**a** raw image

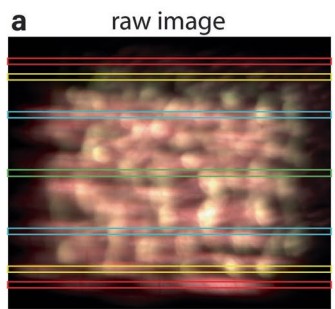

**b** deconvolution

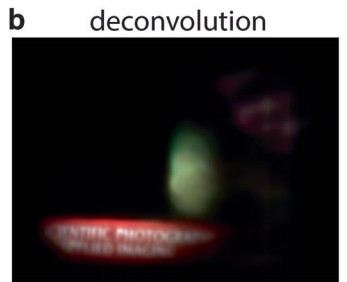

FISTA+anisotropic 3D TV

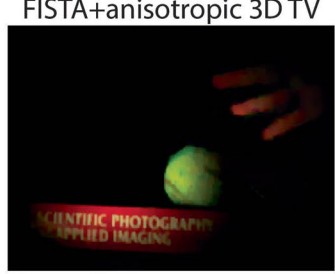

NSTM

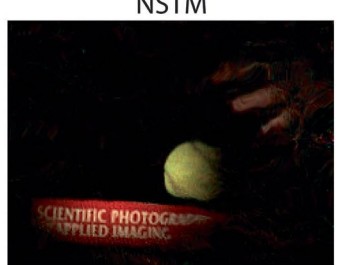

**c**

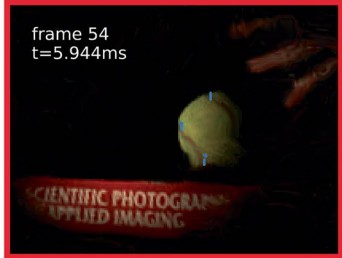

frame 54
t=5.944ms

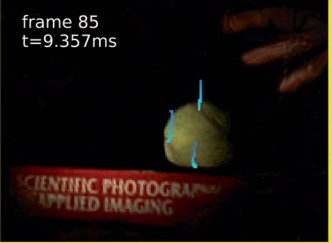

frame 85
t=9.357ms

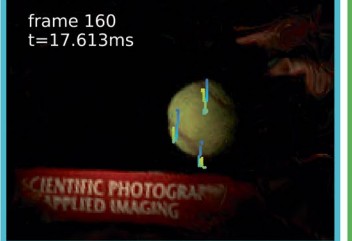

frame 160
t=17.613ms

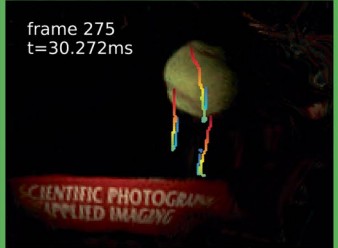

frame 275
t=30.272ms

**Extended Data Fig. 7 | Results for rolling-shutter DiffuserCam. a**, The raw image measurement. **b**, Comparisons of the reconstruction using basic deconvolution (assumes a static scene), FISTA with anisotropic 3D Total Variation regularization (TV)[27] (the original reconstruction method), and our NSTM algorithm. **c**, NSTM reconstruction at different timepoints, with their corresponding measurement rows indicated by colored boxes on the raw image. The colored curves show some selected motion trajectories recovered by the motion network.

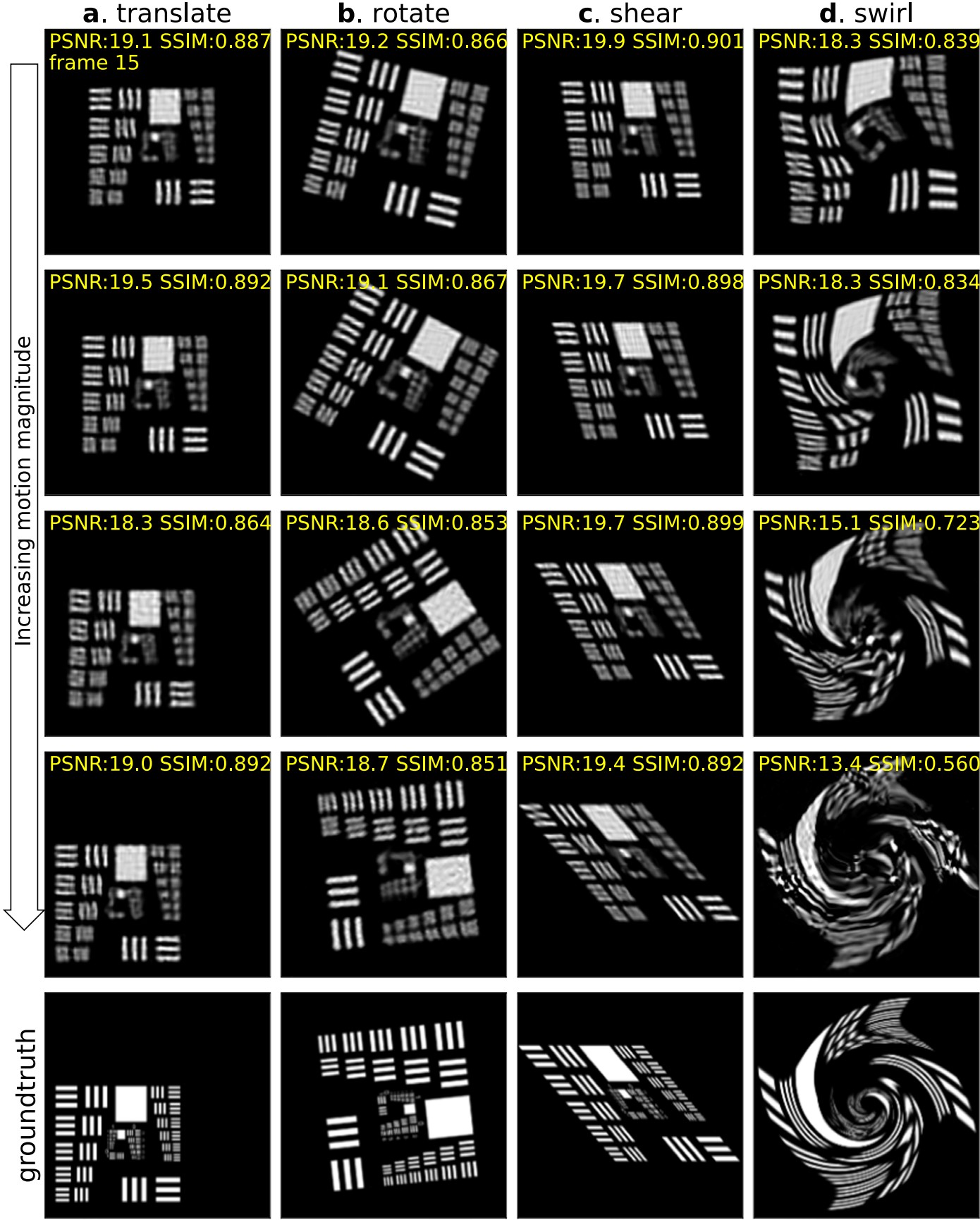

**Extended Data Fig. 8 | SIM simulations with various types and magnitudes of motion.** From left to right: **a**, rigid motion - translation, **b**, rigid motion - rotation, **c**, non-rigid global motion - shearing, and **d**, local deformable motion - swirl. The first four rows show the NSTM reconstructions from simulated images with increasing magnitude of motion between frames, and the last row shows the groundtruth scenes. The reconstruction of local deformable motion is more likely to fail when the motion magnitude increases. Full videos of the dynamic reconstructions can be found in Supplementary Video 7.

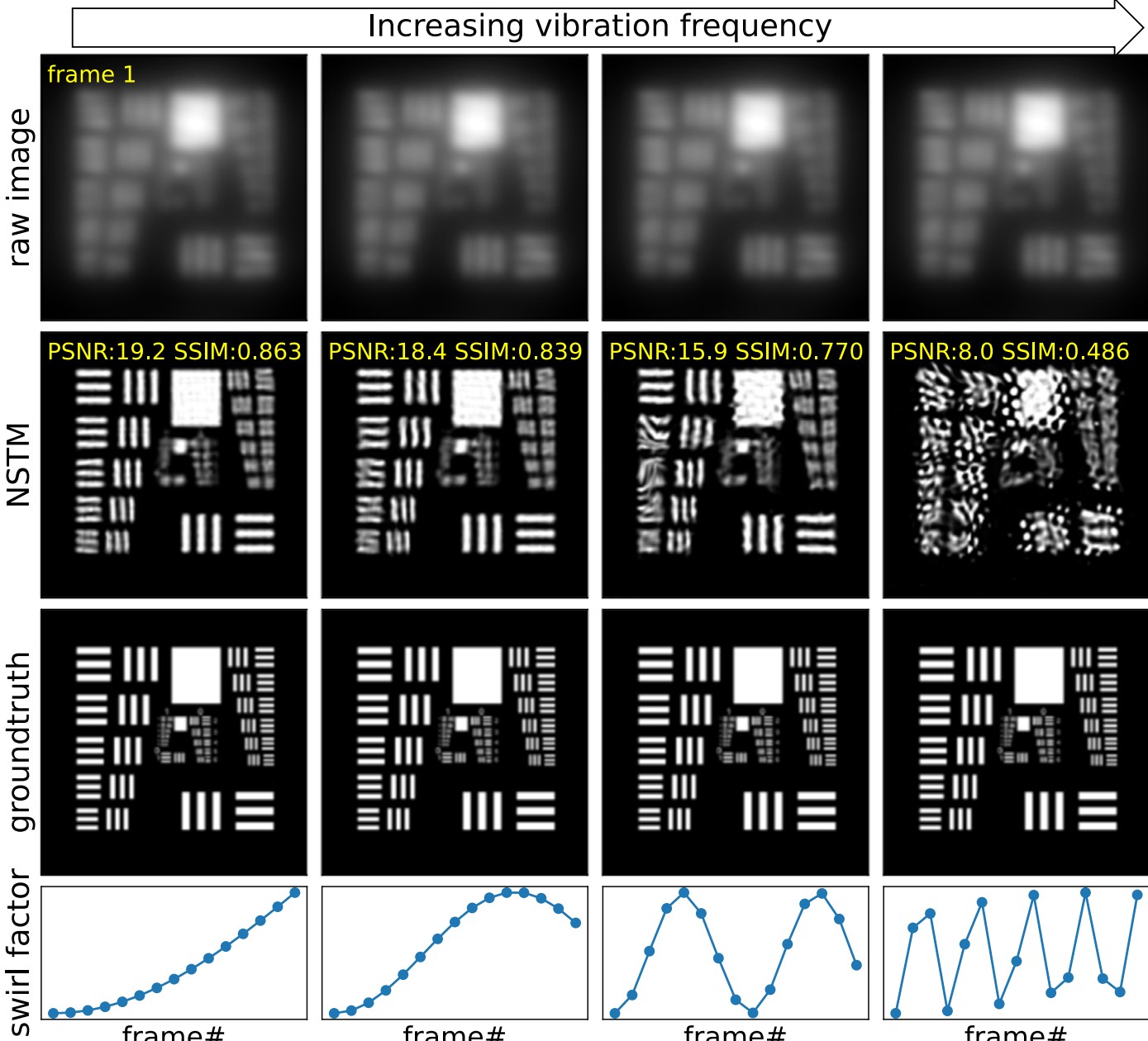

**Extended Data Fig. 9 | Simulations of SIM with local deformable vibration motion.** The deformable swirl motion for each frame is generated using the swirl factor shown in the last row. The frequency of the swirl factor increases from left to right. As the frequency increases, there will be less temporal redundancy between adjacent frames, and hence NSTM will be more likely to fail. Full videos of the dynamic reconstructions can be found in Supplementary Video 8.

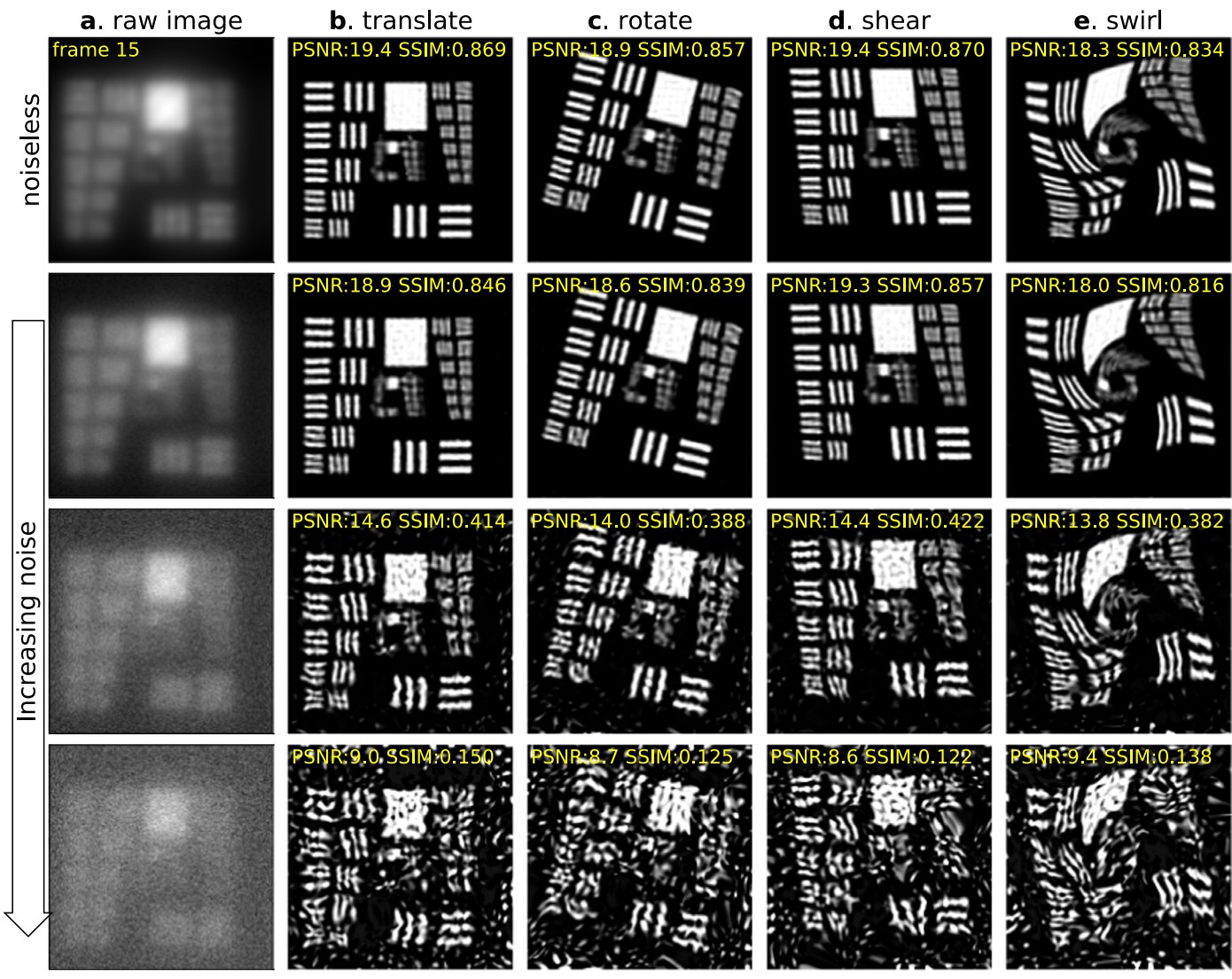

**Extended Data Fig. 10 | Simulations of SIM with increasing amounts of additive Gaussian noise. a**, The simulated raw image. **b**–**e**, Various types of motion: **b**, rigid motion - translation, **c**, rigid motion - rotation, **d**, non-rigid global motion - shearing, and **e**, local deformable motion - swirl. NSTM reconstruction degrades as the noise gets stronger for all types of motion. Full videos of the dynamic reconstructions can be found in Supplementary Video 9.

# Reporting Summary

## Statistics

For all statistical analyses, confirm that the following items are present in the figure legend, table legend, main text, or Methods section.

| n/a | Confirmed | |
|---|---|---|
| ☐ | ☒ | The exact sample size (*n*) for each experimental group/condition, given as a discrete number and unit of measurement |
| ☒ | ☐ | A statement on whether measurements were taken from distinct samples or whether the same sample was measured repeatedly |
| ☒ | ☐ | The statistical test(s) used AND whether they are one- or two-sided *Only common tests should be described solely by name; describe more complex techniques in the Methods section.* |
| ☐ | ☒ | A description of all covariates tested |
| ☐ | ☒ | A description of any assumptions or corrections, such as tests of normality and adjustment for multiple comparisons |
| ☒ | ☐ | A full description of the statistical parameters including central tendency (e.g. means) or other basic estimates (e.g. regression coefficient) AND variation (e.g. standard deviation) or associated estimates of uncertainty (e.g. confidence intervals) |
| ☒ | ☐ | For null hypothesis testing, the test statistic (e.g. *F*, *t*, *r*) with confidence intervals, effect sizes, degrees of freedom and *P* value noted *Give P values as exact values whenever suitable.* |
| ☒ | ☐ | For Bayesian analysis, information on the choice of priors and Markov chain Monte Carlo settings |
| ☒ | ☐ | For hierarchical and complex designs, identification of the appropriate level for tests and full reporting of outcomes |
| ☒ | ☐ | Estimates of effect sizes (e.g. Cohen's *d*, Pearson's *r*), indicating how they were calculated |

*Our web collection on statistics for biologists contains articles on many of the points above.*

## Software and code

Policy information about availability of computer code

| Data collection | Zeiss Zen Black (v14.0.9.201) |
|---|---|
| Data analysis | Our main method was implemented with Python 3.9, jax v0.3.18, flax v0.6.0, CalCIL v0.0.1 (https://github.com/rmcao/CalCIL), nstm v0.1 (https://github.com/rmcao/nstm). Baseline study: cuda-accelerated three-beam 3D SIM reconstruction software (cudasirecon) v1.2.0 (https://github.com/scopetools/cudasirecon), fairSIM v1.5.0 (https://www.fairsim.org). Visualizations were made with matplotlib v3.5.3, napari v0.4.18 (for Supplementary Video 3 and 5). Cell sorting was performed with BD FACSDiva v9.0.2 and FlowJo v10.10.0. |

For manuscripts utilizing custom algorithms or software that are central to the research but not yet described in published literature, software must be made available to editors and reviewers. We strongly encourage code deposition in a community repository (e.g. GitHub). See the Nature Portfolio guidelines for submitting code & software for further information.

## Data

Policy information about availability of data

All manuscripts must include a data availability statement. This statement should provide the following information, where applicable:

- Accession codes, unique identifiers, or web links for publicly available datasets
- A description of any restrictions on data availability
- For clinical datasets or third party data, please ensure that the statement adheres to our policy

Our code is available at https://github.com/rmcao/nstm. SIM datasets collected in this study were deposited in Zenodo at https://doi.org/10.5281/

zenodo.13204660. DPC and rolling-shutter DiffuserCam datasets were obtained from Biomed. Opt. Express 9, 5456-5466 (2018) and https://doi.org/10.1109/ICCPHOT.2019.8747341, and are also available at https://github.com/rmcao/nstm.

## Human research participants

Policy information about studies involving human research participants and Sex and Gender in Research.

| | |
|---|---|
| Reporting on sex and gender | n/a |
| Population characteristics | n/a |
| Recruitment | n/a |
| Ethics oversight | n/a |

Note that full information on the approval of the study protocol must also be provided in the manuscript.

# Field-specific reporting

Please select the one below that is the best fit for your research. If you are not sure, read the appropriate sections before making your selection.

☒ Life sciences      ☐ Behavioural & social sciences      ☐ Ecological, evolutionary & environmental sciences

For a reference copy of the document with all sections, see nature.com/documents/nr-reporting-summary-flat.pdf

# Life sciences study design

All studies must disclose on these points even when the disclosure is negative.

| | |
|---|---|
| Sample size | We chose a diverse set of imaging systems and samples to demonstrate our method. No sample size-based statistics is involved in this study, as this study focus on microscopy reconstruction method and does not have any hypothesis testing. |
| Data exclusions | No data was excluded. |
| Replication | The results of NSTM reconstruction can be replicated using the processing software included in the submission files.<br>The microbead with vibrating motion experiment shown in Fig.2 was independently performed nine times. The optical table was pushed and released in each time. Seven out of nine acquired datasets are suitable for NSTM reconstruction and produce similar results. The remaining two datasets suffer from severe motion blur in individual raw images and thus cannot be recovered by NSTM. |
| Randomization | Neural networks are initialized with random weights, and the dataset is randomly ordered during each epoch of training. Besides, randomization was not relevant to our study, as our study does not have any hypothesis testing. |
| Blinding | Blinding was not relevant to our study as our method was based on computational metrics and algorithms that do not require subjective assessment. |

# Reporting for specific materials, systems and methods

We require information from authors about some types of materials, experimental systems and methods used in many studies. Here, indicate whether each material, system or method listed is relevant to your study. If you are not sure if a list item applies to your research, read the appropriate section before selecting a response.

## Materials & experimental systems

| n/a | Involved in the study |
|---|---|
| ☒ | ☐ Antibodies |
| ☐ | ☒ Eukaryotic cell lines |
| ☒ | ☐ Palaeontology and archaeology |
| ☒ | ☐ Animals and other organisms |
| ☒ | ☐ Clinical data |
| ☒ | ☐ Dual use research of concern |

## Methods

| n/a | Involved in the study |
|---|---|
| ☒ | ☐ ChIP-seq |
| ☐ | ☒ Flow cytometry |
| ☒ | ☐ MRI-based neuroimaging |

# Eukaryotic cell lines

Policy information about <u>cell lines and Sex and Gender in Research</u>

| | |
|---|---|
| Cell line source(s) | The cell line hTERT RPE-1 was obtained from ATCC (https://www.atcc.org/products/crl-4000); gender female. The HEK293t cell line was obtained from ATCC https://www.atcc.org/products/crl-3216; gender female. |
| Authentication | ATCC cell lines arrive with certificate of analysis and obtained from the UC Berkeley Biosciences Division Cell Culture Facility. These lines are distributed commercially. |
| Mycoplasma contamination | The cell lines were tested negative for mycoplasma. |
| Commonly misidentified lines (See <u>ICLAC</u> register) | The RPE-1 and HEK293t cells are not part of commonly misidentified lines. |

# Flow Cytometry

## Plots

Confirm that:

☒ The axis labels state the marker and fluorochrome used (e.g. CD4-FITC).

☒ The axis scales are clearly visible. Include numbers along axes only for bottom left plot of group (a 'group' is an analysis of identical markers).

☒ All plots are contour plots with outliers or pseudocolor plots.

☒ A numerical value for number of cells or percentage (with statistics) is provided.

## Methodology

| | |
|---|---|
| Sample preparation | The RPE-1 cells were transformed using lentivirus transduction as described in the materials and methods, they were suspended using trypsin, and cultured using DMEM F-12 media. The cells were kept on ice at all times during sorting. After sorting they were spun down and resuspended and plated in 10 cm dishes before being used for imaging experiments. |
| Instrument | BD FACSAria (TM) Fusion Cell Sorter (HHMI) was used the in the LSA flow cytometry core at UC Berkeley. |
| Software | BD FACSDiva Software and FlowJo was used to collect and analyze flow cytometry data. |
| Cell population abundance | The cells were approximately 95% (ER) and 78% (mitochondria) StayGold positive respectively after sorting. The halo-tagged F-Actin cells were 99% halo positive after sorting. |
| Gating strategy | The StayGold negative cells were used for gating within live cells using wild-type RPE cells with no fluorophore. This gate was used to designate the StayGold positive cells. Within this population, approximately top 5% of StayGold positive cells were sorted for use in imaging experiments. Gating strategy included in the supplementary figure. The halo negative cells were used for gating within live cells using wild-type RPE cells with no fluorophore. This gate was used to designate the Halo positive cells in the BB515A channel (due to use of the JF503 ligand to gate Halo-positive cells). Within the Halo positive population, top 5% of cells were sorted. Gating strategy is included in supplementary figure. |

☒ Tick this box to confirm that a figure exemplifying the gating strategy is provided in the Supplementary Information.

