## [Peer Review File · Nature Methods]

Peer Review Information

Manuscript Title: Neural space-time model for dynamic multi-shot imaging

Corresponding author name(s): Professor Laura Waller

Editorial Notes: None

Reviewer Comments & Decisions:

Decision Letter, initial version:

9th Mar 2024

Dear Laura,

Your Article, "Neural space-time model for dynamic scene recovery in multi-shot computational imaging systems", has now been seen by three reviewers. As you will see from their comments below, although the reviewers find your work of considerable potential interest, they have raised a number of concerns. We are interested in the possibility of publishing your paper in Nature Methods, but would like to consider your response to these concerns before we reach a final decision on publication.

We therefore invite you to revise your manuscript to address these concerns. The overarching/overlapping concerns of the referees seem to be in regards to pushing the limits of the method and thereby showing its general applicability to different types and scales of motion. We ask that you focus your revision on addressing these concerns. Regarding a napari plugin (ref 3), a napari or ImageJ plugin would be highly desirable, but is not strictly required. At the very least, please make it very clear how new users can access and implement the tool. Regarding computational costs (ref 1), please just make sure these are transparent to readers of the main text, and if the computational burden is high, please discuss how this could be improved in the discussion.

We are committed to providing a fair and constructive peer-review process. Do not hesitate to contact us if there are specific requests from the reviewers that you believe are technically impossible or

unlikely to yield a meaningful outcome.

[REDACTED]

We hope to receive your revised paper within three months. If you cannot send it within this time, please let us know. In this event, we will still be happy to reconsider your paper at a later date so long as nothing similar has been accepted for publication at Nature Methods or published elsewhere.

OPEN SCIENCE REQUIREMENTS

REPORTING SUMMARY AND EDITORIAL POLICY CHECKLISTS

Please note that these forms are dynamic ‘smart pdfs’ and must therefore be downloaded and completed in Adobe Reader. We will then flatten them for ease of use by the reviewers. If you would like to reference the guidance text as you complete the template, please access these flattened versions at <http://www.nature.com/authors/policies/availability.html>.

DATA AVAILABILITY

All novel DNA and RNA sequencing data, protein sequences, genetic polymorphisms, linked genotype and phenotype data, gene expression data, macromolecular structures, and proteomics data must be deposited in a publicly accessible database, and accession codes and associated hyperlinks must be provided in the “Data Availability” section.

Please include a “Data availability” subsection in the Online Methods. This section should inform readers about the availability of the data used to support the conclusions of your study, including accession codes to public repositories, references to source data that may be published alongside the paper, unique identifiers such as URLs to data repository entries, or data set DOIs, and any other statement about data availability. At a minimum, you should include the following statement: “The data that support the findings of this study are available from the corresponding author upon request”, describing which data is available upon request and mentioning any restrictions on availability. If DOIs are provided, please include these in the Reference list (authors, title, publisher (repository name), identifier, year). For more guidance on how to write this section please see: <http://www.nature.com/authors/policies/data/data-availability-statements-data-citations.pdf>

CODE AVAILABILITY

Please include a “Code Availability” subsection in the Online Methods which details how your custom code is made available. Only in rare cases (where code is not central to the main conclusions of the paper) is the statement “available upon request” allowed (and reasons should be specified).

For more information on our code sharing policy and requirements, please see: <https://www.nature.com/nature-research/editorial-policies/reporting-standards#availability-of-computer-code>

MATERIALS AVAILABILITY

ORCID

Nature Methods is committed to improving transparency in authorship. As part of our efforts in this

direction, we are now requesting that all authors identified as 'corresponding author' on published papers create and link their Open Researcher and Contributor Identifier (ORCID) with their account on the Manuscript Tracking System (MTS), prior to acceptance. This applies to primary research papers only. ORCID helps the scientific community achieve unambiguous attribution of all scholarly contributions. You can create and link your ORCID from the home page of the MTS by clicking on 'Modify my Springer Nature account'. For more information please visit please visit www.springernature.com/orcid.

Sincerely,
Rita

Rita Strack, Ph.D.
Senior Editor
Nature Methods

Reviewers' Comments:

Reviewer #1:
Remarks to the Author:
My comments is in the attachment file.

Reviewer #2:
Remarks to the Author:
I would like to apologise to the authors and the editors for the delay in providing this review.

A. Summary of the key results

Cao et al. offer a convincing solution to problems of drift/shift across the dataset for multi-shot computational analysis. In multi-shot imaging (such as SIM or ptychography), common algorithms assume that the sample is static across the whole dataset. Any spatial drift/shift may lead to artefacts in

reconstructions. Here, the authors provide a machine learning approach to correct for these types of data corruption. They build a pipeline, using NSTM, made of a sequence of 2 perceptrons: the first one estimating the motion kernel and the second one understanding the scene. The method is then capable of producing an equivalent reconstruction for each raw measurement. Importantly, the motion of the sample across the dataset can also be recovered. That is a unique feature of the presented approach. The method is demonstrated on Differential Phase Contrast (DPC) microscopy, Structured Illumination Microscopy (SIM) and on DiffuserCam imaging. The data shown show convincing improvement of the NSTM over conventional approaches.

B. Originality and significance: if not novel, please include reference

This work shows clear novelty as far as I am aware since no other solution are available for correcting motion artefacts in such a performant and versatile way. The significance is arguably less strong since the applicability is less ubiquitous and limited to applications where sample motion is significant and the use of multi-shot approach. Although I believe that such situations are probably becoming more and more common with the need to study faster biological processes for instance.

C. Data & methodology: validity of approach, quality of data, quality of presentation

The authors use clear and appropriate methods of validation, clear figures and analyses. The dataset with the beads provides some ground truth to compare to, which is elegant, but the other experimental datasets do not. The authors compare the reconstructions to the wide-field equivalent to show how the movement is correctly recapitulated with NSTM whereas artefacts arise in conventional methods.

D. Appropriate use of statistics and treatment of uncertainties

The authors show a range of dataset with a range of applications. This is sensible and appropriate in my view. The quantification and assessment of artefacts arising from motion is not quantitatively defined but the qualitative assessment provided here is convincing.

E. Conclusions: robustness, validity, reliability

The authors discuss some potential limitations of the method and reiterate the benefits of their approach. As suggested below, I would like to see where the approach fails, in order to determine the range of applicability, is that motion speed? Types of motions?

F. Suggested improvements: experiments, data for possible revision

I would be very interested to see how failure looks like using NSTM and where/how to characterise such failures. I wonder whether there are representative use cases where NSTM does not work appropriately that the authors could provide and discuss.

One interesting case would be to push the speed of the approach to the limit such as with SIM for instance. This is also a very clear case for the value of the method to the general microscopy audience.

G. References: appropriate credit to previous work?

As far as I know, the references are appropriately provided. As mentioned below, I'd however like to see more information about the motivation behind the choice of the NSTM for this, placing it in context and giving some general information about the inner workings of the approach.

H. Clarity and context: lucidity of abstract/summary, appropriateness of abstract, introduction and conclusions

This is where the manuscript is not quite to the level that I would expect for Nature Methods, especially due to the broad readership of the journal. I think that the context of developing and using the methods is not sufficient.

With the provided method description, I find it hard to even follow what is actually optimised during parameter adjustment, or even what is actually fed/used by the networks. This could be due to my lack of clear understanding of NSTM but I believe the general audience would find that reassuring and enriching. I should point that I trust the authors have developed a method that works and performs, since the provided examples and analyses are convincing, but the details of the methods and how to use it are difficult to follow from the manuscript in its current form.

Here are a couple of examples where details are missing:

“The network weights of NSTM are optimized through gradient descent to ensure that the rendered measurements match with the acquired measurements.”

What is actually optimised? What are the loss functions? How are the motion and scene networks differently treated? In other words, how do the authors ensure that the motion network actually produces motion maps?

“Notably, NSTM does not use any data priors or pre-training.”

What is meant here? Do the networks always start from a blank state for each multi-shot dataset that needs reconstruction?

“One limitation of our method is that it hinges on the assumption of temporal redundancy, implying moderate overall motion across measurements and correlatable scenes at adjacent timepoints.”

Can you give examples of situations where this fails? What does the need for temporal redundancy mean in practical terms? What does the user need to ensure to make this assumption valid?

“Despite that the two-network design of NSTM allows an explicit motion model and ensures reconstruction fidelity, it also introduces an additional constraint: since the scene network does not depend on the temporal coordinate, any frame of a dynamic scene has to be obtained by deforming a static reconstruction (from the scene network) with a motion kernel (from the motion network).”

I do not understand that. Yet, this seems fundamental to the working of the approach. How is the forward model of the reconstruction included in the pipeline? In other words, how is the SIM

reconstruction introduced in the pipeline for the SIM dataset? Same for other approaches.

“These two assumptions limit the NSTM’s applicability to fluctuating scenes (such as neuron firing or fluorescence photoactivation) or scenes with appearing/disappearing imaging features. To overcome this limit, future work could modify the NSTM architecture to account for the fluctuation and/or incorporate the time-dependency to the scene network.” Could you please clarify what this means? Especially, I would like for the reader to have clarity on cases where the method should not be used.

“A potential extension for NSTM is the frame interpolation for a high frame-rate reconstruction, which can be achieved by oversampling timepoints during the reconstruction rendering, as demonstrated in Extended Movie 5.”

This seems really exciting, and the supplementary video looks very convincing, and fluid. How do you do that with your networks? How can the time points be interpolated? Again, here it’s my curiosity that drives this question, but I feel that this is a strength that could be further highlighted here. This has potentially high impact on imaging community and is currently overlooked by the community.

Basically, what I would like to see in addition here is a consequent description of how the method was developed and details of how this works/how this is used. Probably a set supplementary notes/more details in introduction would be enough here, describing at least the following:

- How the hash embedding work and retain the information about the input data
- How the forward models are incorporated into the pipeline
- How (at high level) does an NSTM work?
- How the motion network is constrained to provide motion output (loss function on output of the motion network?)
- How the choice of hash embedding, MLPs, and split between motion and scene network was motivated, associated with the relevant references
- How does the network integrate information across time?

The authors provide a lot more details about the similar approach taken in their arXiv paper on Speckle Flow SIM, (<https://arxiv.org/abs/2206.01397>). It would be useful to try and incorporate some of these information into this current manuscript, keeping the readership in mind here.

I would also like to have more information about the optimization process here for each dataset, such as loss functions, epochs.

Minor comments:

“However, because these methods collect less data, they come with additional trade-offs such as reduced resolution or cross-talk.” This is a blanket statement about all the previously mentioned

approaches and seems a little reductive, ignoring the subtle compromises made by each of these approaches. I do not think this is a fair statement, which by the way, is not necessary to motivate the current work as far as I am concerned.

The case about the recovery of the dynamics of the biological system could be made stronger in my view, it's a clear strength of the method and although mentioned, could be exploited demonstrated further. At least, there is a clear need in microscopy imaging to go faster and robustly and accurately recover dynamics. For instance, the simple fact that the method recovers an equivalent reconstruction for each time point in the acquisition, as opposed to a single reconstruction for the whole dataset in the conventional approach, is really powerful from the point of view of temporal resolution, let alone the fact that the time points can be further interpolated. I'd love to hear what the authors imaging could be possible here, discussed in more details in discussion for instance. It is however, just a suggestion here of course.

"To avoid this static assumption and preserve the temporal information, we implement the 3D SIM forward model in real space without band separation, rendering each measurement independently from NSTM's reconstruction at the timepoint that the actual measurement is taken" How is this done?

Romain Laine

Reviewer #3:

Remarks to the Author:

This paper introduces an INR-type neural network to jointly estimates the scene and its motion dynamics in order to achieve high-quality computational image reconstruction. The idea is to leverage a variant of SIREN to learn (1) the motion or displacement field of the scene depending on T and xy-coordinates and (2) the reconstruction of the scene after taking the estimated motion into account. To me, the overall method is very inspiring. I really had an "aha moment" when reading the paper and I am totally convinced that SIREN-like method will provide a nice solution to motion estimation and scene reconstruction.

Another thing I really appreciate in the paper is the discussion of limitation (I do have some suggestions on additional discussion regarding limitation at the end). The flip side of SIREN is the requirement of long training per image, which may get worse when dealing with large images or images with complicated contents. The paper did a very good discussion on this issue, which is very important, in my opinion.

Before full endorsement for publication, I do have some concerns and suggestion as below.

(1) I think it would be important to quantitatively demonstrate the limitation of NSTM in terms of handling large displacement or motion. As pointed out by the authors, NSTM assumes the temporal redundancy. Additional experiments showing "how much redundancy" is necessary for NSTM to achieve reasonable results would be critical for users to understand the limitation in a more concrete way and therefore can prepare their own imaging experiments accordingly (e.g., in order to match the required "redundancy").

(2) The scene reconstruction from NSTM should have data range very different from the original acquisition (e.g., converting original 16-bit 0-65535 into float32 -Inf. to Inf.). To me, this could be a caveat for potential users who may care about the intensity of the measurement.

(3) I think the overall idea of using INR-based method for motion estimation and image reconstruction shares significant similarity with INR-based method for image registration. I don't think there is any problem with novelty. I just would suggest to mention INR-based registration either as prior works or related works. see additional references at the very end (disclaimer: I am NOT an author on any of the papers).

(4) another suggestion for better dissemination is to wrap the tool into a napari plugin.

Overall, I think the paper is of very good quality in general, with minor things to be improved before publication.

Additional references:

Byra, M., Poon, C., Rachmadi, M.F. et al. Exploring the performance of implicit neural representations for brain image registration. *Sci Rep* 13, 17334 (2023). <https://doi.org/10.1038/s41598-023-44517-5>

Wolterink, J.M., Zwienenberg, J.C., & Brune, C. (2022). Implicit Neural Representations for Deformable Image Registration. *International Conference on Medical Imaging with Deep Learning*.

Author Rebuttal to Initial comments

Point-to-point response for NMETH-A54634

We would like to thank the editor and reviewers for their constructive comments and suggestions that have helped improve the quality of this manuscript. The manuscript has undergone a thorough revision according to those comments. Please see below for our point-by-point responses.

Response to the editor

Associate Editor Comment — The overarching/overlapping concerns of the referees seem to be in regards to pushing the limits of the method and thereby showing its general applicability to different types and scales of motion. We ask that you focus your revision on addressing these concerns. Regarding a napari plugin (ref 3), a napari or ImageJ plugin would be highly desirable, but is not strictly required. At the very least, please make it very clear how new users can access and implement the tool. Regarding computational costs (ref 1), please just make sure these are transparent to readers of the main text, and if the computational burden is high, please discuss how this could be improved in the discussion.

Reply: Thank you for your constructive feedback. We have addressed those concerns in the revision and improved the quality of the manuscript. Here is a summary of major changes in the revised manuscript:

- We performed a comprehensive study to better understand and illustrate the limit of our method (detailed in Discussion - second paragraph, Extended Fig.9-11 and Extended Movie 6-8).
- We acquired additional 3D SIM experimental data with F-Actin filaments and mitochondrial cristae to demonstrate our method's capacity with denser objects (Extended Figs 6 and 7).
- Following the reviewer's comment, we optimized our 3D SIM reconstruction pipeline and improved the computational efficiency by roughly 20× (Online Methods - 3D SIM reconstruction), and we added a runtime benchmark to report the reconstruction time for each method and computing hardware (Extended Table 1).
- We provided detailed instructions to setup and use our software through multiple examples and interactive visualization through Jupyter notebook.
- We improved the writing of the manuscript and added more descriptions of the model design, reconstruction pipeline, as well as the implementation for each use case in the main text and Online Methods.

Response to the reviewers

Reviewer 1

Thank you for your constructive feedback!

Reviewer Comment 1.1 — Reconstruction in computational imaging systems usually aims to obtain finer results with higher resolution. The proposed NSTM is designed to learn high-quality reconstructions through implicit neural representations with its loss calculated between the forward-model-rendered measurements and the acquired measurements. This learning process is similar to some deconvolution-based optimization algorithm [1][2], which has been known of generating results with inferior resolution than the ground truths, e.g., conventional SIM images. Although the authors have demonstrated the output resolution of NSTM by reconstructing ER and mitochondrial matrix images in 3D-SIM experiments, these two biological structures are relatively simple. Therefore, I doubt whether NSTM is able to reconstruct fine structures with high fidelity for some more complex specimens, e.g., interlaced F-actin filaments and delicate mitochondrial inner cristae, etc.

Reply: Thank you for your constructive comments. We want to note that there are two key differences between the gradient update used by NSTM and the deconvolution-based algorithms used for blind-SIM. First, the two papers mentioned by the reviewer assume that the illumination patterns are initially unknown and need to also estimate the illumination patterns in the reconstruction, which requires more measurements and makes the reconstruction complicated. Second, blind-SIM assumes speckle-like patterns, which reduce the contrast of the super-resolved features when comparing with sinusoidal SIM. We believe it is the unknown speckle illumination, not the gradient update itself, that causes the inferior resolution in those two papers. In fact, many image deconvolution techniques use gradient updates and obtain very high quality images (Ulyanov et al., 2018; Antipa et al., 2018).

Following recommendations of the reviewer, we acquired additional 3D SIM experimental data with F-actin filaments and mitochondrial cristae, with results shown in Extended Figs. 6 and 7. NSTM is able to successfully recover fine structures with motion.

Reviewer Comment 1.2 — In this manuscript, the authors mainly demonstrated the removal of relatively large motion artifact resulted from the movement of massive structures of specimens. However, in most bioimaging experiments, there's another type of motion artifact that arises from fast but subtle vibration of local structures, e.g., the spiky motion artifacts of ER in SIM imaging (see figure below, cropped from paper [3]). Can NSTM be applied to remove such kind of motion artifact?

Reply: No, we don't believe NSTM, at least in its current form, can resolve the subtle high-frequency vibrational motion, like those multi-scale motion exhibited in ER (Nixon-Abell et al., 2016). Such high-frequency vibration would violate the temporal redundancy assumption. We added a simulation study to show how NSTM starts to fail as the motion gets less smooth (Extended Fig. 10). This could be a direction for future research to better model for periodic motion, and we also added this into our discussion.

Reviewer Comment 1.3 — It takes about 10 minutes to reconstruct a 2D-SIM image, and 3 hours for reconstructing a 3D-SIM stack with 8 Nvidia A100 GPUs. However, in most biology labs or even AI labs don't have such vast computing power. Are 8 Nvidia A100 GPUs or equivalent computing resources necessary for 3D-SIM reconstruction? If yes, what are the major limitations of using other low-level GPUs? memory or convergence time, or other factors? If no, what is the least requirement of GPU memory or computing power to reconstruct a 3D-SIM stack, e.g., $512 \times 512 \times 20$ and $1024 \times 1024 \times 20$ after up-sampling? In my opinion, this is an important factor if it will impede the wide applicability of the proposed methods in the community.

Reply: Compute efficiency is indeed the current limiting factor for self-supervised methods like NSTM, as the network is optimized from scratch for each set of raw images. However, we want to stress that there are considerable efforts to optimize its efficiency, from both ourselves and the engineering community. After we submitted the manuscript in last December, we optimized the reconstruction workflow on 3D SIM and improved its speed by around $20\times$ (Online Methods - 3D SIM reconstruction). With this improvement, the NSTM reconstruction of a $20 \times 512 \times 512$ volume takes 40 minutes on a Nvidia A6000 GPU (which costs around \$5000 to buy or \$2 per hour on AWS). We also added a more accurate runtime benchmark for each method and computing hardware to improve the transparency (Extended Table 1). Besides, we are excited to see that the community is moving toward more efficient floating point arithmetic, such as half-precision or even 8-bit floating point. Together with the new computing hardware driven by the rapid development of foundation models, we envision that the efficiency can be further improved by an order of magnitude the next couple of years, which will hopefully remove the computing barrier for a widely applicability. In its current form, we think that the compute requirements are already not infeasible for basic labs doing biology research, which are increasingly relying on compute for other aspects of their research anyways.

Reviewer Comment 1.4 — The manuscript and method section is sort of abbreviated, so it is kind of difficult for me (or other potential readers) to fully understand the implementation of NSTM. More details should be added in Methods or supplementary materials. For example, there should be at least formulated forward model and loss functions for each application scenario of NSTM. In fact, I understood the concept, workflow, and implementation details mostly by reading the provided codes (with few annotation), however, which is fairly time-consuming.

Reply: We can appreciate the request for more details and agree that readers should not need to understand the code in order to understand the method. To address this, we added more detailed descriptions of the workflow and implementation to the manuscript (**Introduction** and **Online Methods**), including the mathematical expressions of the network construction and the optimization process for each application. It is encouraging to hear that the code itself is explanatory, as we spent quite some effort making the code clear and adaptable for other research. We will add more annotations to it in the code release.

Reviewer Comment 1.5 — For the 3D-SIM application, I wonder if the NSTM model has to be trained for each specific timepoint (with 3 orientations * 5 phases) or can be applied onto multiple timepoints? Can the NSTM do temporal interpolation between orientations?

Reply: Yes, NSTM can be trained using images from multiple timepoints. Because of the gradient-based reconstruction used by NSTM, the number of raw images fed into the reconstruction is flexible. After the reconstruction, NSTM can do temporal interpolation simply by feeding arbitrary timepoints

into the network. Compared with the direct interpolation in the pixel space, NSTM is able to perform interpolation in the motion space using the motion network. Extended Movie 5 shows an example of frame interpolation, and we extended its description in **Discussion**.

Reviewer Comment 1.6 — In Fig. 1b, there should be a schematic of the loss function calculation, and the propagation should be derived from the loss rather than the “reconstructed scene”.

Reply: Thanks for catching this; we updated the figure accordingly.

Reviewer Comment 1.7 — Considering the extreme time complexity of NSTM, can the author provide a quantitative evaluation on processing time and memory requirement of NSTM to reconstruct data volumes of different sizes with a certain type of GPU?

Reply: Yes, we added Extended Table 1 and reported the runtime information for each of the various examples we had under three different GPUs.

Reviewer 2

Thank you for your constructive feedback!

Reviewer Comment 2.1 — The authors discuss some potential limitations of the method and reiterate the benefits of their approach. As suggested below, I would like to see where the approach fails, in order to determine the range of applicability, is that motion speed? Types of motions?

I would be very interested to see how failure looks like using NSTM and where/how to characterise such failures. I wonder whether there are representative use cases where NSTM does not work appropriately that the authors could provide and discuss.

Reply: This is an interesting question, though difficult to answer concretely. Assuming a known forward model without model mismatch, NSTM reconstruction success depends on a few factors: type of motion, motion magnitude, and imaging noise. We have added a comprehensive study to illustrate some failure modes in **Discussion** - second paragraph, Extended Fig.9-11 and Extended Movie 6-8. In short, when the motion is linear, the magnitude of motion does not significantly affect the reconstruction; as the motion gets more complex and non-linear (e.g., more local distortions), then greater magnitude of motion will degrade the joint reconstruction. More details can be found in the newly added paragraph.

Reviewer Comment 2.2 — One interesting case would be to push the speed of the approach to the limit such as with SIM for instance. This is also a very clear case for the value of the method to the general microscopy audience.

Reply: Our current 3D SIM images were acquired from a commercial Zeiss microscope, which unfortunately has long delays due to the mechanical components. Thus, its temporal resolution is limited by the hardware delay, not the exposure time, so we weren't able to push this beyond the limits as requested. However, NSTM itself does not limit to the particular hardware we used and can be used on faster systems. We added a simulation result for the scenario where the frame rate is limited by exposure time, such that pushing to a even faster frame rate means lowering the raw images' SNR.

In Extended Fig.11, we show that NSTM is robust to a reasonable amount of noise. We expect that experimental studies would give similar conclusions.

Reviewer Comment 2.3 — As far as I know, the references are appropriately provided. As mentioned below, I'd however like to see more information about the motivation behind the choice of the NSTM for this, placing it in context and giving some general information about the inner workings of the approach.

This is where the manuscript is not quite to the level that I would expect for Nature Methods, especially due to the broad readership of the journal. I think that the context of developing and using the methods is not sufficient.

With the provided method description, I find it hard to even follow what is actually optimised during parameter adjustment, or even what is actually fed/used by the networks. This could be due to my lack of clear understanding of NSTM but I believe the general audience would find that reassuring and enriching. I should point that I trust the authors have developed a method that works and performs, since the provided examples and analyses are convincing, but the details of the methods and how to use it are difficult to follow from the manuscript in its current form.

Reply: We agree with this criticism and have extended the descriptions of methods throughout the revised manuscript. In particular, we added two sections in Online Methods with detailed descriptions and mathematical expressions of how NSTM is constructed (Online Methods - Construction of NSTM) and how NSTM is optimized (Online Methods - NSTM reconstruction). We added more detailed descriptions of the reconstruction workflow and the incorporation of each particular imaging system (Online Methods).

Reviewer Comment 2.4 — “The network weights of NSTM are optimized through gradient descent to ensure that the rendered measurements match with the acquired measurements.” What is actually optimised? What are the loss functions? How are the motion and scene networks differently treated? In other words, how do the authors ensure that the motion network actually produces motion maps?

Reply: The weights of motion and scene networks are optimized using a loss function that consists of the mean square error between the rendered measurements and the actual measurements. The motion and scene networks are treated identically, except that we found a smaller learning rate for motion network to be helpful for a stable convergence. The two-network construction of NSTM ensures a meaningful reconstruction of motion, since the scene network only takes spatial coordinates (but not temporal coordinates) as its input, and the optimization will converge only when the motion network recovers the motion map faithfully. These information can be found in the newly added sections Online Methods - Construction of NSTM and NSTM reconstruction.

Reviewer Comment 2.5 — “Notably, NSTM does not use any data priors or pre-training.” What is meant here? Do the networks always start from a blank state for each multi-shot dataset that needs reconstruction?

Reply: Yes exactly. We added some clarification to this in the text: “Notably, NSTM does not use any data priors or pre-training, such that the network weights are trained from scratch for each set of raw measurements.”

Reviewer Comment 2.6 — “One limitation of our method is that it hinges on the assumption of temporal redundancy, implying moderate overall motion across measurements and correlatable scenes at adjacent timepoints.” Can you give examples of situations where this fails? What does the need for temporal redundancy mean in practical terms? What does the user need to ensure to make this assumption valid?

Reply: This comment is similar to Comment 2.1, and so was addressed above. In practical terms, NSTM uses the two-network construction to model for a dynamic scene which has some temporal redundancy, such that any frame of a dynamic scene has to be obtained by deforming a static reconstruction (from the scene network) with a motion kernel (from the motion network). We also rephrased this paragraph to make this more clear.

Reviewer Comment 2.7 — “Despite that the two-network design of NSTM allows an explicit motion model and ensures reconstruction fidelity, it also introduces an additional constraint: since the scene network does not depend on the temporal coordinate, any frame of a dynamic scene has to be obtained by deforming a static reconstruction (from the scene network) with a motion kernel (from the motion network).” I do not understand that. Yet, this seems fundamental to the working of the approach. How is the forward model of the reconstruction included in the pipeline? In other words, how is the SIM reconstruction introduced in the pipeline for the SIM dataset? Same for other approaches.

Reply: We apologize for the confusion. Yes, this is fundamental to NSTM, which should be able to resolve the previous comment. We added a section in Online Methods to better describe the construction of NSTM (Online Methods - Construction of NSTM) and also another section to describe the reconstruction pipeline (Online Methods - NSTM reconstruction). The idea is that every frame should contain the same parts of the object as the previous frame, just with things moving around or deforming over time, in order to fit with our model’s two-network construction.

Reviewer Comment 2.8 — “These two assumptions limit the NSTM’s applicability to fluctuating scenes (such as neuron firing or fluorescence photoactivation) or scenes with appearing/disappearing imaging features. To overcome this limit, future work could modify the NSTM architecture to account for the fluctuation and/or incorporate the time-dependency to the scene network.” Could you please clarify what this means? Especially, I would like for the reader to have clarity on cases where the method should not be used.

Reply: This is related to the previous comment. If parts of the scene being imaged are disappearing and/or appearing over time, rather than just moving around within the frame (as is the case with neurons turning ‘on’ and ‘off’ over time), then it won’t fit with our model and the reconstruction may fail. As in Comment 2.1, we added a thorough discussion of failure modes in the second paragraph of Discussion and also rephrased this paragraph to clarify the limitations.

Reviewer Comment 2.9 — “A potential extension for NSTM is the frame interpolation for a high frame-rate reconstruction, which can be achieved by oversampling timepoints during the reconstruction rendering, as demonstrated in Extended Movie 5.” This seems really exciting, and the supplementary video looks very convincing, and fluid. How do you do that with your networks? How can the time points be interpolated? Again, here it’s my curiosity that drives this question,

but I feel that this is a strength that could be further highlighted here. This has potentially high impact on imaging community and is currently overlooked by the community.

Reply: Thank you for your comment and interest. Once the NSTM has been optimized for a set of raw images, this temporal interpolation can be achieved by simply feeding intermediate timepoints to NSTM for the interpolated frames. However, we want to note that there is generally no guarantee about the accuracy of these interpolated frames since we have did not take any measurements at those intermediate timepoints. We have also extended its description in the fifth paragraph of **Discussion** to clarify this.

Reviewer Comment 2.10 — Basically, what I would like to see in addition here is a consequent description of how the method was developed and details of how this works/how this is used. Probably a set supplementary notes/more details in introduction would be enough here, describing at least the following:

- How the hash embedding work and retain the information about the input data
- How the forward models are incorporated into the pipeline
- How (at high level) does an NSTM work?
- How the motion network is constrained to provide motion output (loss function on output of the motion network?)
- How the choice of hash embedding, MLPs, and split between motion and scene network was motivated, associated with the relevant references
- How does the network integrate information across time?

The authors provide a lot more details about the similar approach taken in their arXiv paper on Speckle Flow SIM, (<https://arxiv.org/abs/2206.01397>). It would be useful to try and incorporate some of these information into this current manuscript, keeping the readership in mind here.

I would also like to have more information about the optimization process here for each dataset, such as loss functions, epochs.

“To avoid this static assumption and preserve the temporal information, we implement the 3D SIM forward model in real space without band separation, rendering each measurement independently from NSTM’s reconstruction at the timepoint that the actual measurement is taken” How is this done?

Reply: Thank you for your comment. We have added more descriptions in the main text, Online Methods, and Appendix and addressed these issues accordingly. Point-to-point responses below:

- We added more detailed descriptions of hash embedding in **Online Methods - Construction of NSTM** and **Appendix A**. The purpose of hash embedding is not to retain input information, but to map the input into a higher dimensional space and expand the capacity of the coordinate-based network.
- We expanded the descriptions of the reconstruction pipeline in **Online Methods** and included a detailed description of the forward model incorporation for each imaging system.
- We put more comprehensive descriptions about the working mechanism of NSTM in the main text and **Online Methods**, and we added a new section in **Online Methods** to better describe the design of NSTM.

- The motion network does not use a separate loss term, but it is instead guided by the gradient back-propagated from the scene network and the MSE loss. Please refer to Comment 2.4 for a more detailed response.
- The choice of hash embedding and MLP and their associated references can be found in **Online Methods - Construction of NSTM**. The split design of motion and scene networks is now elaborated in the fourth paragraph of **Introduction**.
- The temporal information is accounted by the motion network which provides a motion displacement map to generate a dynamic scene from a time-independent scene network. We added more detailed descriptions of the reconstruction pipeline in the newly added section **Online Methods - NSTM reconstruction**.
- We added descriptions of loss function and optimization settings in **Online Methods - NSTM reconstruction**, and the number of epochs can be found in the newly added Extended Table 1.
- We added a very detailed description of the forward model we used for SIM in **Online Methods - 3D SIM reconstruction**.

Reviewer Comment 2.11 — “However, because these methods collect less data, they come with additional trade-offs such as reduced resolution or cross-talk.” This is a blanket statement about all the previously mentioned approaches and seems a little reductive, ignoring the subtle compromises made by each of these approaches. I do not think this is a fair statement, which by the way, is not necessary to motivate the current work as far as I am concerned.

Reply: We agree with you and rephrased this sentence accordingly.

Reviewer Comment 2.12 — The case about the recovery of the dynamics of the biological system could be made stronger in my view, it's a clear strength of the method and although mentioned, could be exploited demonstrated further. At least, there is a clear need in microscopy imaging to go faster and robustly and accurately recover dynamics. For instance, the simple fact that the method recovers an equivalent reconstruction for each time point in the acquisition, as opposed to a single reconstruction for the whole dataset in the conventional approach, is really powerful from the point of view of temporal resolution, let alone the fact that the time points can be further interpolated. I'd love to hear what the authors imaging could be possible here, discussed in more details in discussion for instance. It is however, just a suggestion here of course.

Reply: We're glad the reviewers appreciate the benefits of finer-resolution temporal dynamic reconstructions. We agree that this method could enable the study of fast biological dynamics and this is a key contribution. We tried our best to demonstrate the robustness of NSTM across different scales and highlight the high-quality reconstruction at frame rates on par with the speed of each collected image (not each collected dataset). In the revised manuscript, we have promoted this benefit to make it more prominent. We also spend quite some effort making sure that our code is well-written and the software is accessible and easy to use by users. We hope to see new applications from real biologists to realize its full potential.

Reviewer 3

Thank you for your constructive feedback!

Reviewer Comment 3.1 — I think it would be important to quantitatively demonstrate the limitation of NSTM in terms of handling large displacement or motion. As pointed out by the authors, NSTM assumes the temporal redundancy. Additional experiments showing "how much redundancy" is necessary for NSTM to achieve reasonable results would be critical for users to understand the limitation in a more concrete way and therefore can prepare their own imaging experiments accordingly (e.g., in order to match the required "redundancy").

Reply: Showing the limits/failure of the method for large motion was brought up by multiple reviewers, and is addressed above in the previous reviewer comment reply (Comment 2.1). We included new results showing how and when the method fails in a number of motion cases, but we unfortunately haven't found a good metric to assess the temporal redundancy quantitatively. To summarize our newly added results, assuming a known forward model without model mismatch, we found that NSTM reconstruction depends on a few factors: type of motion, motion magnitude (speed), forward model and imaging noise. When the motion is linear, the magnitude of motion does not significantly affect the reconstruction; as the motion gets more complex and non-linear (e.g., more local distortions), then greater magnitude of motion will degrade the joint reconstruction. More details can be found in the newly added paragraph (**Discussion** - second paragraph), Extended Fig.9-11 and Extended Movie 6-8.

Reviewer Comment 3.2 — The scene reconstruction from NSTM should have data range very different from the original acquisition (e.g., converting original 16-bit 0-65535 into float32 -Inf. to Inf.). To me, this could be a caveat for potential users who may care about the intensity of the measurement.

Reply: This is a good point, and a rather general issue for computational image reconstruction. In fact, the conventional reconstructions for DPC, 3D SIM and rolling-shutter DiffuserCam will all experience this issue, so we believe adding NSTM to the reconstruction will not make it more limiting if users opt to use these multi-shot methods in the first place.

Reviewer Comment 3.3 — I think the overall idea of using INR-based method for motion estimation and image reconstruction shares significant similarity with INR-based method for image registration. I don't think there is any problem with novelty. I just would suggest to mention INR-based registration either as prior works or related works. see additional references at the very end (disclaimer: I am NOT an author on any of the papers).

Byra, M., Poon, C., Rachmadi, M.F. et al. Exploring the performance of implicit neural representations for brain image registration. *Sci Rep* 13, 17334 (2023). <https://doi.org/10.1038/s41598-023-44517-5>

Wolterink, J.M., Zwienenberg, J.C., & Brune, C. (2022). Implicit Neural Representations for Deformable Image Registration. *International Conference on Medical Imaging with Deep Learning*.

Reply: Thank you for the references; they are very relevant and have been cited in the revised version of the manuscript. We agreed that NSTM and these two registration methods have some similarity in terms of they both use INR type of networks. However, we found that our task of multi-shot reconstruction and single-modal registration are inherently very different, since 1) the object is not known in NSTM

beforehand, which makes the motion estimation significantly more challenging, and 2) different raw images may have different contrast in our case, which makes NSTM closer to multi-modal registration (versus the existing papers are all single modal). We would also like to highlight that our project was started in early 2021, directly inspired by (Mildenhall et al., 2021; Sitzmann et al., 2020), and we have presented the work-in-progress in various places since 2022. For these reasons, we do not consider them as prior work and will mention them as related work.

Reviewer Comment 3.4 — Another suggestion for better dissemination is to wrap to tool into a napari plugin.

Reply: Thank you for your suggestion. We agree that a napari or imagej plugin would be very helpful to improve the accessibility of our method. However, at the current stage, we expect users to run the algorithm in dedicated GPU servers because of the significant computation requirement, while napari or imagej does not have good support for server’s side deployment so far to our knowledge. Instead, we provide a well-written software package with a step-by-step setup instructions and examples for different use cases. We provide both python scripts for mass deployment and Jupyter notebooks for interactive visualization. In the next month or two when we publish the code, we will migrate our Jupyter notebooks to Google Colab platform, so that users without GPU server access could also easily test our software.

References

- Antipa, N., Kuo, G., Heckel, R., Mildenhall, B., Bostan, E., Ng, R., and Waller, L. (2018). Dif-fusercam: lensless single-exposure 3d imaging. *Optica*, 5(1):1–9.
- Mildenhall, B., Srinivasan, P. P., Tancik, M., Barron, J. T., Ramamoorthi, R., and Ng, R. (2021). Nerf: Representing scenes as neural radiance fields for view synthesis. *Communications of the ACM*, 65(1):99–106.
- Nixon-Abell, J., Obara, C. J., Weigel, A. V., Li, D., Legant, W. R., Xu, C. S., Pasolli, H. A., Harvey, K., Hess, H. F., Betzig, E., et al. (2016). Increased spatiotemporal resolution reveals highly dynamic dense tubular matrices in the peripheral er. *Science*, 354(6311):aaf3928.
- Sitzmann, V., Martel, J., Bergman, A., Lindell, D., and Wetzstein, G. (2020). Implicit neural representations with periodic activation functions. *Advances in neural information processing systems*, 33:7462–7473.
- Ulyanov, D., Vedaldi, A., and Lempitsky, V. (2018). Deep image prior. In *Proceedings of the IEEE conference on computer vision and pattern recognition*, pages 9446–9454.

Decision Letter, first revision:

20th May 2024

Dear Laura,

Thank you for your letter detailing how you would respond to the reviewer concerns regarding your Article, "Neural space-time model for dynamic multi-shot imaging". We have decided to invite you to revise your manuscript as you have outlined, before we reach a final decision on publication. We think the SIM data in the paper are already of sufficient quality and do not require additional data. We ask that you update the software as described for a final quick check by reviewer 3. Please do update the manuscript to discuss relevant points raised by reviewer 2.

[REDACTED]

We hope to receive your revised paper within 4-6 weeks. If you cannot send it within this time, please let us know. In this event, we will still be happy to reconsider your paper at a later date so long as nothing similar has been accepted for publication at Nature Methods or published elsewhere.

OPEN SCIENCE REQUIREMENTS

REPORTING SUMMARY AND EDITORIAL POLICY CHECKLISTS

Please note that these forms are dynamic ‘smart pdfs’ and must therefore be downloaded and completed in Adobe Reader. We will then flatten them for ease of use by the reviewers. If you would like to reference the guidance text as you complete the template, please access these flattened versions at <http://www.nature.com/authors/policies/availability.html>.

DATA AVAILABILITY

Please include a “Data availability” subsection in the Online Methods. This section should inform readers about the availability of the data used to support the conclusions of your study, including accession codes to public repositories, references to source data that may be published alongside the paper, unique identifiers such as URLs to data repository entries, or data set DOIs, and any other statement about data availability. At a minimum, you should include the following statement: “The data that support the findings of this study are available from the corresponding author upon request”, describing which data is available upon request and mentioning any restrictions on availability. If DOIs are

provided, please include these in the Reference list (authors, title, publisher (repository name), identifier, year). For more guidance on how to write this section please see:
<http://www.nature.com/authors/policies/data/data-availability-statements-data-citations.pdf>

CODE AVAILABILITY

Please include a “Code Availability” subsection in the Online Methods which details how your custom code is made available. Only in rare cases (where code is not central to the main conclusions of the paper) is the statement “available upon request” allowed (and reasons should be specified).

For more information on our code sharing policy and requirements, please see:
<https://www.nature.com/nature-research/editorial-policies/reporting-standards#availability-of-computer-code>

MATERIALS AVAILABILITY

ORCID

Nature Methods is committed to improving transparency in authorship. As part of our efforts in this direction, we are now requesting that all authors identified as ‘corresponding author’ on published papers create and link their Open Researcher and Contributor Identifier (ORCID) with their account on the Manuscript Tracking System (MTS), prior to acceptance. This applies to primary research papers only. ORCID helps the scientific community achieve unambiguous attribution of all scholarly contributions. You can create and link your ORCID from the home page of the MTS by clicking on ‘Modify my Springer Nature account’. For more information please visit please

visit www.springernature.com/orcid.

Sincerely,
Rita

Rita Strack, Ph.D.
Senior Editor
Nature Methods

Reviewers' Comments:

Reviewer #1:

Remarks to the Author:

The authors have considerably improved the manuscript, especially the additions in illustrating the forward model, loss functions, training details, etc., make it easier for potential reader to understand the implementation of NSTM. And I agree with author that the efficiency of NSTM will be optimized with the efforts of the entire community and the authors themselves.

However, I'm not fully convinced by the results shown in new Extended Fig. 6 and 7 that NSTM is competent in temporal interpolation between orientations and SIM timepoints (as claimed in Reply 1.5) and removing the reconstruction artifacts and for relatively complex biological structures. The reasons are as below:

- The 3D-SIM image reconstructed by either the conventional algorithm or NSTM presented massive reconstruction artifacts.
- The image shown in Extended Fig. 7 looks like outer mitochondrial membrane more than mitochondrial cristae (but it was labeled by LifeAct-Halo according to the figure caption?)
- The motion between orientation or SIM timepoints was too small to illustrate the temporal interpolation capability of NSTM. Since mitochondria has been recognized as a very dynamic organelles, the cells seem to be less active than normal status.

I strongly suggest the authors to 1) improve the sample preparation or imaging configuration to optimize the overall 3D-SIM imaging quality, and 2) add two new videos to show the temporal

interpolation and artifact elimination capability.

Reviewer #2:

Remarks to the Author:

The authors have satisfactorily addressed my points in the revised manuscript.

Reviewer #3:

Remarks to the Author:

In this revision, the authors address all my concerns. Thanks for the great effort!

However, I noticed one issue I didn't pay attention to. I didn't look at the code in my initial review as I was not asked to check the code. This time, I just had a quick look at the code and the readme file. I realized that the current documentation was mostly focusing on "how to reproduce the results shown in the paper". However, from a user's point of view, it would be more important to add at least some information on what to do when a user needs to use this on their own data. Just to name one, when I opened the jupyter notebook as suggested in the readme, I see quite a few hard-coded parameters, which I assume need to be tuned if users need to use this software for their own data.

Author Rebuttal, first revision:

Point-to-point response for NMETH-A54634A

We would like to thank the editor and reviewers again for their constructive comments and suggestions.

Response to the editor

Associate Editor Comment — We have decided to invite you to revise your manuscript as you have outlined, before we reach a final decision on publication. We think the SIM data in the paper are already of sufficient quality and do not require additional data. We ask that you update the software as described for a final quick check by reviewer 3. Please do update the manuscript to discuss relevant points raised by reviewer 2.

Reply: In this revision, we have updated the software package as outlined in the revision plan. In particular, we included comprehensive documentation for the usage and adaptation of our software as well as two demos on Google Colab. We think the updated code package would sufficiently resolve the issue raised by Reviewer 3. After the code is made public, we will strive to continue improving it based on the community's feedback and making it more accessible and easier to use.

To address the points raised by Reviewer 1, we updated the Extended Fig. 6 and added a video to better illustrate the result shown in Extended Fig. 6.

Response to the reviewers

Reviewer 1

Reviewer Comment 1.1 — I'm not fully convinced by the results shown in new Extended Fig. 6 and 7 that NSTM is competent in temporal interpolation between orientations and SIM timepoints (as claimed in Reply 1.5) and removing the reconstruction artifacts and for relatively complex biological structures. The reasons are as below:

The 3D-SIM image reconstructed by either the conventional algorithm or NSTM presented massive reconstruction artifacts.

Reply: Eliminating reconstruction artifacts originating from the shifted DC components (background signal) and from the sample induced distortion (scattering) is not the focus of this work. The reconstruction quality is ultimately limited by imaging hardware and biological reagents. Rather, our work is effective to eliminate (within reasonable bounds) hallucinations/blurred reconstruction as a consequence of sample features moving during acquisition, as demonstrated in Extended Fig 6 and the newly added Extended Movie 5.

Reviewer Comment 1.2 — The image shown in Extended Fig. 7 looks like outer mitochondrial membrane more than mitochondrial cristae (but it was labeled by LifeAct-Halo according to the figure caption?)

Reply: The figure caption was indeed incorrect. The cell line was stably expressing LifeAct-Halo, however, it was not stained with a Halo ligand, instead it was stained with MitoTracker as described in the Sample Preparation. We have since removed this supplemental figure as other examples shown in this paper illustrate the motion correction (see below).

Reviewer Comment 1.3 — The motion between orientation or SIM timepoints was too small to illustrate the temporal interpolation capability of NSTM. Since mitochondria has been recognized as a very dynamic organelles, the cells seem to be less active than normal status.

Reply: It is true there is not much motion in this particular dataset of the cell stained by MitoTracker, but our capability to perform motion-resolved reconstruction was demonstrated throughout this paper. We have tested our method with structures such as mito (Fig 3 & Extended Fig 4), ER (Fig 4 & Extended Fig 5) and F-actin (Extended Fig 6). We believe that we have enough examples in the rest of the paper, and thus we removed this supplementary figure of MitoTracker to avoid confusion.

Reviewer Comment 1.4 — I strongly suggest the authors to 1) improve the sample preparation or imaging configuration to optimize the overall 3D-SIM imaging quality, and 2) add two new videos to show the temporal interpolation and artifact elimination capability.

Reply: These experimental data are representative for users likely to use 3D-SIM with access to commercial hardware and reagents. We included several examples in our manuscript to convince readers our motion resolving capability. Following your recommendation, we made additional video visualization to showcase our dynamic reconstruction for F-actin data in Extended Fig. 6.

Reviewer 2

Reviewer 3

Reviewer Comment 3.1 — I noticed one issue I didn't pay attention to. I didn't look at the code in my initial review as I was not asked to check the code. This time, I just had a quick look at the code and the readme file. I realized that the current documentation was mostly focusing on "how to reproduce the results shown in the paper". However, from a user's point of view, it would be more important to add at least some information on what to do when a user needs to use this on their own data. Just to name one, when I opened the jupyter notebook as suggested in the readme, I see quite a few hard-coded parameters, which I assume need to be tuned if users need to use this software for their own data.

Reply: Thank you for your constructive feedback! We have been improving the code performance since submitting the manuscript for review. In the previous code, the imaging system's parameters and model parameters are stored in .yaml config files, but the notebook did have some hard-coded parameters that should be moved to the config files. In the revised code, we have greatly simplified model parameters and removed unused parameters. Besides, we now included all pre-processing scripts, from raw data to reconstruction. We made a detailed API documentation as well as a guide to use your own data and/or different forward models (under docs/_build/html/index.html). In addition, we also made Google Colab demos whose links can be found in the README file.

Decision Letter, second revision:

18th Jun 2024

Dear Laura,

Thank you for submitting your revised manuscript "Neural space-time model for dynamic multi-shot imaging" (N METH-A54634B). It has now been seen again by reviewer 3 and their comments are below. The reviewers find that the paper/code has improved in revision, and therefore we'll be happy in principle to publish it in Nature Methods, pending minor revisions to comply with our editorial and formatting guidelines.

TRANSPARENT PEER REVIEW

Please note: we allow redactions to authors' rebuttal and reviewer comments in the interest of confidentiality. If you are concerned about the release of confidential data, please let us know specifically what information you would like to have removed. Please note that we cannot incorporate redactions for any other reasons. Reviewer names will be published in the peer review files if the reviewer signed the comments to authors, or if reviewers explicitly agree to release their name. For more information, please refer to our FAQ page.

ORCID

Sincerely,
Rita

Rita Strack, Ph.D.
Senior Editor
Nature Methods

Reviewer #3 (Remarks to the Author):

I think all issues have been addressed.

Reviewer #3 (Remarks on code availability):

looks good now

Final Decision Letter:

Dear Laura,

I am pleased to inform you that your Article, "Neural space-time model for dynamic multi-shot imaging", has now been accepted for publication in Nature Methods. The received and accepted dates will be Dec 2, 2023 and August 15, 2024. This note is intended to let you know what to expect from us over the next month or so, and to let you know where to address any further questions.

Over the next few weeks, your paper will be copyedited to ensure that it conforms to Nature Methods style. Once your paper is typeset, you will receive an email with a link to choose the appropriate publishing options for your paper and our Author Services team will be in touch regarding any additional information that may be required. It is extremely important that you let us know now whether you will be difficult to contact over the next month. If this is the case, we ask that you send us the contact information (email, phone and fax) of someone who will be able to check the proofs and deal with any last-minute problems.

Please note that *Nature Methods* is a Transformative Journal (TJ). Authors may publish their research with us through the traditional subscription access route or make their paper immediately open access through payment of an article-processing charge (APC). Authors will not be required to make a final decision about access to their article until it has been accepted. Find out more about Transformative Journals

If you are active on Twitter/X, please e-mail me your and your coauthors' handles so that we may tag you when the paper is published.

Best regards,
Rita

Rita Strack, Ph.D.
Senior Editor
Nature Methods